# Comparator-Adaptive $\Phi$-Regret: Improved Bounds, Simpler Algorithms, and Applications to Games

**Soumita Hait**[*]
University of Southern California
hait@usc.edu

**Ping Li**[*]
Shanghai University of Finance and Economics
pinglee@stu.sufe.edu.cn

**Haipeng Luo**[*]
University of Southern California
haipengl@usc.edu

**Mengxiao Zhang**[*]
University of Iowa
mengxiao-zhang@uiowa.edu

## Abstract

In the classic expert problem, $\Phi$-regret measures the gap between the learner's total loss and that achieved by applying the best action transformation $\phi \in \Phi$. A recent work by Lu et al. [2025] introduces an adaptive algorithm whose regret against a comparator $\phi$ depends on a certain sparsity-based complexity measure of $\phi$, (almost) recovering and interpolating optimal bounds for standard regret notions such as external, internal, and swap regret. In this work, we propose a general idea to achieve an even better comparator-adaptive $\Phi$-regret bound via much simpler algorithms compared to Lu et al. [2025]. Specifically, we discover a prior distribution over all possible binary transformations and show that it suffices to achieve prior-dependent regret against these transformations. Then, we propose two concrete and efficient algorithms to achieve so, where the first one learns over multiple copies of a prior-aware variant of the Kernelized MWU algorithm of Farina et al. [2022b], and the second one learns over multiple copies of a prior-aware variant of the BM-reduction [Blum and Mansour, 2007]. To further showcase the power of our methods and the advantages over [Lu et al., 2025] besides the simplicity and better regret bounds, we also show that our second approach can be extended to the game setting to achieve accelerated and adaptive convergence rate to $\Phi$-equilibria for a class of general-sum games. When specified to the special case of correlated equilibria, our bound improves over the existing ones from Anagnostides et al. [2022a,b].

## 1 Introduction

Expert problem [Freund and Schapire, 1997] is one of the most fundamental online learning problems, where a learner repeatedly hedges over $d$ experts with the goal of being comparative to a strong benchmark. More concretely, in each round $t$, the learner proposes a distribution $p_t \in \Delta(d)$ over $d$ experts and suffers loss $\langle p_t, \ell_t \rangle$ where $\ell_t \in [0,1]^d$ is a loss vector decided by an adversary. Consider a benchmark that always applies a fixed linear transformation $\phi : \Delta(d) \mapsto \Delta(d)$ to the learner's strategy and thus suffers loss $\langle \phi(p_t), \ell_t \rangle$ in round $t$. The regret of the learner against $\phi$ is then defined as $\text{Reg}(\phi) \triangleq \sum_{t=1}^{T} \langle p_t - \phi(p_t), \ell_t \rangle$, that is, the difference between the learner's total loss and that of the benchmark. Given a class of linear transformations $\Phi$, the learner's $\Phi$-regret is defined as $\max_{\phi \in \Phi} \text{Reg}(\phi)$ [Greenwald and Jafari, 2003]. With an appropriate choice of $\Phi$, this general notion of $\Phi$-regret subsumes many well-studied regret notions in the literature, such as external regret, internal regret, and swap regret.

---

[*]Authors are listed in alphabetical order.

While the optimal $\Phi$-regret bound naturally depends on the complexity of the class $\Phi$ and different algorithms have been proposed for different $\Phi$'s in the literature, a recent work by Lu et al. [2025] developed a comparator-adaptive algorithm whose regret against $\phi$ depends on a certain sparsity-based complexity measure $c_\phi$ of $\phi$, almost recovering the optimal regret bounds for external regret, internal regret, and swap regret simultaneously via one single algorithm. Specifically, their algorithm achieves $\mathrm{Reg}\,(\phi) = \mathcal{O}\Big(\sqrt{c_\phi(T+d)\,(\log d)^3}\Big)$ for all $\phi$ simultaneously, where $c_\phi = \min\{d - d_\phi^{\mathrm{self}}, d - d_\phi^{\mathrm{unif}} + 1\}$, $d_\phi^{\mathrm{self}}$ is the number of experts that are mapped to themselves by $\phi$, and $d_\phi^{\mathrm{unif}}$ is the maximum number of experts mapped to the same expert by $\phi$ (see Section 2 for formal definitions). The design of their algorithm, however, is somewhat complicated and uses Haar-wavelet-inspired matrix features.

In this work, we significantly improve over Lu et al. [2025] by *developing simpler algorithms, achieving better comparator-adaptive regret bounds, and demonstrating broader applications to accelerated convergence in games*. Specifically, our contributions are as follows.

- First, in Section 3, we propose a general idea of achieving an improved comparator-adaptive regret bound $\mathrm{Reg}(\phi) = \mathcal{O}(\sqrt{c_\phi T \log d})$, removing both the extra $\widetilde{\mathcal{O}}(\sqrt{c_\phi d})$ additive term and also the extra $\log d$ factor compared to that of Lu et al. [2025]. We achieve so by proposing a prior distribution $\pi$ over all binary and linear transformations and showing that as long as a natural prior-dependent regret bound $\mathrm{Reg}(\phi) = \mathcal{O}(\sqrt{T \log(1/\pi(\phi))})$ holds, then the aforementioned new comparator-adaptive regret bound holds.

- While at first glance it is unclear at all how to achieve the prior-dependent regret bound above efficiently (since the number of all binary transformations is $d^d$), we propose two efficient approaches to achieve so thanks to the special structure of our prior. For the first approach (Section 4), we utilize and extend the Kernelized Multiplicative Weight Update algorithm of Farina et al. [2022b] and show that a certain prior-dependent kernel can be computed efficiently; for the second approach (Section 5), we develop a prior-aware variant of the classic BM-reduction [Blum and Mansour, 2007] and learn over multiple copies of it. Both approaches are arguably much simpler than the algorithm of Lu et al. [2025].

- Besides its simplicity and better regret bounds, we further demonstrate the power of our second approach by extending it to an uncoupled learning dynamic for games and achieving accelerated and adaptive convergence to $\Phi$-equilibria (Section 6). Specifically, we develop an algorithm such that, when deployed by all players for a broad class of $N$-player general-sum games considered by Anagnostides et al. [2022c], each player enjoys a $T$-independent regret bound $\mathrm{Reg}(\phi) = \mathcal{O}(c_\phi N \log d + N^2 \log d)$ for all $\phi$ simultaneously. Based on standard connection between $\Phi$-regret and $\Phi$-equilibria, this implies an adaptive $(\max_{\phi \in \Phi} c_\phi N \log d + N^2 \log d)/T$ convergence rate to $\Phi$-equilibria, simultaneously for all classes $\Phi$, which is the first result of this kind to our knowledge. Moreover, when specified to the case of correlated equilibria (where $\Phi$ is all binary linear transformations), we improve over Anagnostides et al. [2022b] on the $d$-dependence and remove any $\mathrm{polylog}(T)$ dependence compared to Anagnostides et al. [2022a,b] (although their results hold more generally for any general-sum games). Our technique is also new and relies on the flexibility and a particular structure of our second approach, which allows us to bound the path-length of the learning dynamic via showing small external regret. We remark that it is highly unclear (if possible at all) how to achieve similar results using the algorithm of Lu et al. [2025] (or even our first approach).

**Related Work**   We refer the reader to Cesa-Bianchi and Lugosi [2006] for detailed discussions on external regret (e.g., [Freund and Schapire, 1997]), internal regret (e.g., [Foster and Vohra, 1999, Stoltz and Lugosi, 2005]), and swap regret [Blum and Mansour, 2007], whose formal definition can be found in Section 2. As mentioned, they all belong to the family of $\Phi$-regret, a concept proposed by Greenwald and Jafari [2003] and further studied in many subsequent works such as Stoltz and Lugosi [2007], Gordon et al. [2008], Rakhlin et al. [2011], Piliouras et al. [2022], Bernasconi et al. [2023], Cai et al. [2024], Zhang et al. [2024] due to its generality and connection to various equilibrium concepts. However, comparator-adaptive $\Phi$-regret bounds were only recently considered by Lu et al. [2025] as far as we know.

The concept of comparator-adaptive regret, nevertheless, is much older and has been studied under various different contexts; we refer the reader to Orabona [2019] for in-depth discussion. The algorithm of Lu et al. [2025] makes use of advances from this line of work [Cutkosky, 2018],

while ours uses two simpler ideas: prior-dependent external regret via the classic Multiplicative Weight Update (MWU) algorithm [Littlestone and Warmuth, 1994, Freund and Schapire, 1997] and combining multiple algorithms to learn over the learning rates via a meta MWU (an idea that has been used in many prior works such as Koolen et al. [2014], Van Erven and Koolen [2016], Foster et al. [2017], Cutkosky [2019], Bhaskara et al. [2020], Chen et al. [2021]).

The connection between online learning and games dates back to Blackwell [1956], Hannan [1957], Freund and Schapire [1999]. Greenwald and Jafari [2003] showed that in a general-sum game, if all players deploy an online learning algorithm with sublinear $\Phi$-regret, then the empirical distribution of their joint strategy profiles converges to a $\Phi$-equilibrium with the convergence rate being the average (over time) $\Phi$-regret. While $\Phi$-regret is usually of order $\sqrt{T}$ in the worst case (leading to $1/\sqrt{T}$ convergence rate), since the work of Daskalakis et al. [2011], Rakhlin and Sridharan [2013], Syrgkanis et al. [2015], there has been a surge of research showing that accelerated convergence rate of order $\mathrm{polylog}(T)/T$ is possible in many cases by utilizing the structure of the game and certain optimistic online learning algorithms [Daskalakis et al., 2021, Anagnostides et al., 2022a,b, Farina et al., 2022a]. Our result in Section 6 adds to the growing body of this line of work and is the first accelerated convergence rate that is also adaptive in the complexity of $\Phi$. Our approach also makes use of standard optimistic online learning algorithms, but existing analysis does not work directly due to various technical hurdles. We resolve them by exploiting a particular structure of our second algorithm, borrowing ideas from a two-layer framework of Zhang et al. [2022], and considering a subclass of games where the sum of all players' external regret is always nonnegative (a broad class as shown by Anagnostides et al. [2022c]).

## 2   Preliminaries

**General Notations**   For a positive integer $n$, let $[n]$ denote the set $\{1, 2, \ldots, n\}$. Define $\mathbb{R}_+^n$ to be the positive orthant of the $n$-dimensional Euclidean space, and $\Delta(n) \triangleq \{p \in \mathbb{R}_+^n, \sum_{i=1}^n p_i = 1\}$ to be the $(n-1)$-dimensional simplex. Given a finite set $S$, denote $|S|$ to be its cardinality and $\Delta(S)$ to be the set of probability distributions over $S$. Given $p, q \in \Delta(n)$, define $\mathrm{KL}(p, q) \triangleq \sum_{i=1}^n p_i \log \frac{p_i}{q_i}$ as the KL-divergence between $p$ and $q$. For a matrix $M \in \mathbb{R}^{m \times n}$, we denote by $M_{i:} \in \mathbb{R}^n$ the $i$-th row of $M$ and $M_{:j} \in \mathbb{R}^m$ the $j$-th column of $M$. For two matrices $M_1, M_2 \in \mathbb{R}^{m \times n}$, define the inner product $\langle M_1, M_2 \rangle \triangleq \mathrm{trace}\left(M_1^\top M_2\right)$. Let $\mathbf{1}$ and $\mathbf{0}$ be the all-one and all-zero vector in an appropriate dimension, let $e_i$ be the one-hot vector in an appropriate dimension with the $i$-th entry being 1 and all other entries being 0, and let $\mathbf{I}$ be the identity matrix in an appropriate dimension.

Define $\mathcal{S} \triangleq \{\phi \in [0, 1]^{d \times d} \mid \phi_{k:} \in \Delta(d), \forall\, k \in [d]\}$ as the set of all row-stochastic matrices, which is also the set of all possible linear transformations from $\Delta(d)$ to $\Delta(d)$ if we treat each $\phi \in \mathcal{S}$ as a linear operator: $\phi(p) = \phi^\top p$. The subset $\Phi_b \triangleq \{\phi \in \{0, 1\}^{d \times d} \mid \phi_{k:} \in \Delta(d), \forall\, k \in [d]\} \subseteq \mathcal{S}$ consisting of all binary row-stochastic matrices is of particular interest. For a distribution $\pi \in \Delta(\Phi_b)$, we let $\pi(\phi)$ be the probability mass of $\phi \in \Phi_b$.

**Expert Problem and $\Phi$-regret**   In an expert problem, the interaction between the environment and the learner proceeds for $T$ rounds. At each round $t \in [T]$, the learner decides a distribution $p_t \in \Delta(d)$ over the $d$ experts and the environment decides a loss vector $\ell_t \in [0, 1]^d$. The learner then receives $\ell_t$ and suffers loss $\langle p_t, \ell_t \rangle$. Given a transformation $\phi \in \mathcal{S}$, the regret of the learner against this $\phi$ is defined as $\mathrm{Reg}(\phi) \triangleq \sum_{t=1}^T \langle p_t - \phi(p_t), \ell_t \rangle$, and given a class of transformations $\Phi \subseteq \mathcal{S}$, the $\Phi$-regret is defined as $\mathrm{Reg}(\Phi) \triangleq \max_{\phi \in \Phi} \mathrm{Reg}(\phi)$ [Greenwald and Jafari, 2003].

With an appropriate choice of $\Phi$, $\Phi$-regret reduces to many standard regret notions. For example, with $\Phi = \Phi_{\mathsf{Ext}} \triangleq \{\mathbf{1}e_i^\top\}_{i \in [d]}$, $\Phi$-regret recovers the standard *external regret* that competes with a fixed expert, and it is well known that the minimax bound in this case is $\Theta(\sqrt{T \log d})$, achieved by for example the classic Multiplicative Weight Update (MWU) algorithm [Littlestone and Warmuth, 1994, Freund and Schapire, 1997]; with $\Phi = \Phi_{\mathsf{Int}} \triangleq \{\mathbf{I} - e_i e_i^\top + e_i e_j^\top\}_{i,j \in [d], i \neq j}$, $\Phi$-regret recovers *internal regret* and competes with a strategy that moves all the weights for expert $i$ to expert $j$ for some fixed $i$ and $j$, and the minimax bound in this case is also $\Theta(\sqrt{T \log d})$ [Stoltz and Lugosi, 2005]; and with $\Phi = \Phi_b$, $\Phi$-regret reduces to *swap regret* and competes with all possible swaps

between experts, and the minimax bound in this case is $\Theta(\sqrt{dT \log d})$ [Blum and Mansour, 2007, Ito, 2020] for a certain regime of $T$ and $d$.[2]

In a recent work by Lu et al. [2025], they derive a comparator-adaptive regret bound of the form $\text{Reg}(\phi) = \mathcal{O}\left(\sqrt{c_\phi (T + d) (\log d)^3}\right)$ for all $\phi \in \mathcal{S}$ simultaneously, where $c_\phi$ is a certain sparsity-based complexity measure of $\phi$, formally defined as follows.

**Definition 2.1** (Complexity measure of $\phi$ from Lu et al. [2025]). *For any $\phi \in \Phi_b$, define $c_\phi \triangleq \min\{d - d_\phi^{\text{self}}, d - d_\phi^{\text{unif}} + 1\}$, where $d_\phi^{\text{self}}$, the degree of self-map of $\phi$, is the number of experts $i$ such that $\phi(e_i) = e_i$ (equivalently, $d_\phi^{\text{self}} = \text{trace}(\phi)$), and $d_\phi^{\text{unif}}$, the degree of uniformity of $\phi$, is the multiplicity of the most frequent element in the multi-set $\{\phi(e_1), \ldots, \phi(e_d)\}$. For any $\phi \in \mathcal{S} \setminus \Phi_b$, define $c_\phi \triangleq \min_{q \in Q_\phi} \mathbb{E}_{\phi' \sim q}[c_{\phi'}]$ where $Q_\phi = \{q \in \Delta(\Phi_b) : \mathbb{E}_{\phi' \sim q}[\phi'] = \phi\}$.*

Direct calculation shows that $\max_{\phi \in \Phi_{\text{Ext}}} c_\phi = \max_{\phi \in \Phi_{\text{Int}}} c_\phi = 1$ and $\max_{\phi \in \Phi_b} c_\phi = d$, and thus their algorithm achieves almost optimal external regret, internal regret, and swap regret simultaneously.

We remark that, in fact, Lu et al. [2025] define $c_\phi$ for $\phi \in \mathcal{S} \setminus \Phi_b$ using the exact same definition as the case when $\phi \in \Phi_b$, which is rather unnatural and results in a discontinuous function over $\mathcal{S}$ — for example, a slight perturbation for a $\phi \in \Phi_b$ with a large $d_\phi^{\text{self}}$ (and thus small $c_\phi$) can lead to a $\phi' \in \mathcal{S}$ with $d_{\phi'}^{\text{self}} = 0$ (and thus potentially large $c_{\phi'}$). The definition we use here, on the other hand, is a continuous and natural extension from $\Phi_b$ to $\mathcal{S}$. It can be shown that our definition leads to a strictly smaller complexity measure; see Proposition A.1 in Appendix A for more discussion.

However, what is perhaps not realized by Lu et al. [2025] is that their bound $\text{Reg}(\phi) = \mathcal{O}\left(\sqrt{c_\phi (T + d) (\log d)^3}\right)$ in fact also holds under our better definition of $c_\phi$ (via the same algorithm). The reasoning is the same as our proof for Theorem 3.3: it suffices to show this bound for $\phi \in \Phi_b$ and then take convex combination of the bound when dealing with $\phi \in \mathcal{S} \setminus \Phi_b$. This explains why we change the definition to this version.

One may wonder why we care about any $\phi \in \mathcal{S} \setminus \Phi_b$ — after all, since the benchmark $\sum_{t=1}^T \langle \phi(p_t), \ell_t \rangle$ is linear in $\phi$, the best $\phi$ from a set $\Phi$ is always on its boundary. The reason is that what the learner ultimately cares about is her total loss $\sum_{t=1}^T \langle p_t, \ell_t \rangle = \sum_{t=1}^T \langle \phi(p_t), \ell_t \rangle + \text{Reg}(\phi)$, and when a comparator-adaptive bound on $\text{Reg}(\phi)$ is available, we should consider the $\phi$ that minimizes the sum $\sum_{t=1}^T \langle \phi(p_t), \ell_t \rangle + \text{Reg}(\phi)$, instead of just $\sum_{t=1}^T \langle \phi(p_t), \ell_t \rangle$, and in this case, it is totally possible that the best $\phi$ is not in $\Phi_b$.

# 3   Achieving $c_\phi$-Dependent Regret via a Special Prior

In this section, we present a new and general idea to achieve a $c_\phi$-dependent bound for $\text{Reg}(\phi)$. To this end, we define a prior distribution $\pi$ over $\Phi_b$ through the following definitions, which plays an important role in our approach.

**Definition 3.1** ($\psi$-induced distribution). *Given a row-stochastic matrix $\psi \in \mathcal{S}$, it induces a distribution $\pi_\psi \in \Delta(\Phi_b)$ such that $\pi_\psi(\phi) = \Pi_{i \in [d]} \langle \psi_{i:}, \phi_{i:} \rangle$ for all $\phi \in \Phi_b$.[3]*

**Definition 3.2** (special prior distribution $\pi$). *The prior distribution $\pi$ over $\Phi_b$ is a mixture of $d + 1$ distributions such that*

$$\pi \triangleq \frac{1}{2d} \sum_{k=1}^d \pi_{\psi^k} + \frac{1}{2} \pi_{\psi^{d+1}}, \tag{1}$$

*where $\psi^1, \ldots, \psi^{d+1} \in \mathcal{S}$ are defined as*

$$\psi^k \triangleq \frac{d-2}{d-1} \cdot \mathbf{1} e_k^\top + \frac{1}{d(d-1)} \mathbf{1}\mathbf{1}^\top, \forall k \in [d], \text{ and } \psi^{d+1} \triangleq \frac{d-2}{d-1} \cdot \mathbf{I} + \frac{1}{d(d-1)} \mathbf{1}\mathbf{1}^\top. \tag{2}$$

---

[2]More concretely, for the regime where $d \log d \lesssim T \lesssim d^{3/2}/(\log d)$. For other regimes, see recent work by Dagan et al. [2024], Peng and Rubinstein [2024].

[3]We remark that this is a valid distribution since $\sum_{\phi \in \Phi_b} \pi_\psi(\phi) = \sum_{\phi \in \Phi_b} \prod_{i,j \in [d]: \phi_{ij}=1} \psi_{ij} = \prod_{i=1}^d \sum_{j=1}^d \psi_{ij} = \prod_{i=1}^d 1 = 1$.

It is straightforward to verify that $\psi^1, \ldots, \psi^{d+1}$ are indeed row-stochastic matrices. In fact, when viewed as transformation rules, each $\psi^k$ (for $k \in [d]$) transforms all experts to expert $k$ with a large probability mass of $1 - 1/d$ and to other experts uniformly with the remaining mass, and similarly, $\psi^{d+1}$ transforms each expert to itself with a large probability mass of $1 - 1/d$ and to other experts uniformly with the remaining mass. At a high-level, $\psi^{d+1}$ is intuitively connected to $d_\phi^{\text{self}}$ in the definition of $c_\phi$ and $\{\psi^k\}_{k \in [d]}$ are connected to $d_\phi^{\text{unif}}$. Building on such connections, we prove the following main result.

**Theorem 3.3.** *For any $\phi \in \Phi_b$, we have $\log(\frac{1}{\pi(\phi)}) \leq 2 + 2c_\phi \log d$. Consequently, if an algorithm achieves*

$$\text{Reg}(\phi) = \mathcal{O}\left(\sqrt{T \log\left(\frac{1}{\pi(\phi)}\right)} + B\right) \tag{3}$$

*for all $\phi \in \Phi_b$ and some $\phi$-independent term $B$, then it also achieves $\text{Reg}(\phi) = \mathcal{O}\left(\sqrt{(1 + c_\phi \log d)T} + B\right)$ for all $\phi \in \mathcal{S}$ simultaneously.*

We defer the proof to Appendix A and give some intuition here by considering two special cases. First, consider a $\phi \in \Phi_{\text{Ext}}$: we know that $\phi = \mathbf{1} e_i^\top$ for some $i \in [d]$ and thus $\pi(\phi) \geq \frac{1}{2d} \pi_{\psi^i}(\phi) = \frac{1}{2d}(1 - \frac{1}{d})^d = \Theta(1/d)$, meaning that $\log(1/\pi(\phi))$ is of order $\log d$ and consistent with $c_\phi \log d$. As another example, consider a $\phi \in \Phi_{\text{Int}}$: we have $\phi = \mathbf{I} - e_i e_i^\top + e_i e_j^\top$ for some $i \neq j$ and thus $\pi(\phi) \geq \frac{1}{2} \pi_{\psi^{d+1}}(\phi) = \frac{1}{2}(1 - \frac{1}{d})^{d-1} \cdot \frac{1}{d(d-1)} = \Theta(1/d^2)$, which means $\log(1/\pi(\phi))$ is also of order $\log d$ and consistent with $c_\phi \log d$.

To see why Eq. (3) is a natural bound one should aim for, we recall a standard idea from Blum and Mansour [2007], Gordon et al. [2008] that reduces the $\Phi$-regret for the expert problem to the standard (external) regret of an Online Linear Optimization (OLO) problem over $\Phi$: if at each round $t$, the proposed distribution over experts $p_t \in \Delta(d)$ is computed as the stationary distribution of some $\phi_t \in \mathcal{S}$ (that is, $p_t = \phi_t(p_t)$), then we have $\text{Reg}(\phi) = \sum_{t=1}^T \langle p_t - \phi(p_t), \ell_t \rangle = \sum_{t=1}^T \langle \phi_t(p_t) - \phi(p_t), \ell_t \rangle = \sum_{t=1}^T \langle \phi_t - \phi, p_t \ell_t^\top \rangle$, which means $\text{Reg}(\phi)$ is exactly the standard regret of the sequence $\phi_1, \ldots, \phi_T$ against a fixed $\phi$ for an OLO instance with $\langle \cdot, p_t \ell_t^\top \rangle$ as the linear loss function in round $t$. We can solve this OLO instance by treating it as yet another expert problem with $\Phi_b$ as the expert set, in which case a bound in the form of Eq. (3) is just the standard prior-dependent regret achievable by many algorithms, such as MWU.

The caveat, of course, is that naively doing so is computationally inefficient since the size of $\Phi_b$ is $d^d$. In fact, a similar concern was raised by Lu et al. [2025] as a motivation for their totally different approach. However, thanks to the special structure of our prior $\pi$, we manage to develop two different efficient approaches to achieve Eq. (3), as shown in the next two sections.

**Regret comparison with Lu et al. [2025]**    In our two approaches that achieve Eq. (3), the term $B$ is either $\mathcal{O}(\sqrt{T \log \log d})$ or $\mathcal{O}(\sqrt{T \log d})$, making our final regret bound essentially $\mathcal{O}(\sqrt{c_\phi T \log d})$. Compared to the bound $\mathcal{O}\left(\sqrt{c_\phi(T + d)(\log d)^3}\right)$ of Lu et al. [2025], we have thus removed the extra $\widetilde{\mathcal{O}}(\sqrt{c_\phi d})$ additive term and also the extra $\log d$ factor. When specified to standard regret notations (external/internal/swap regret), our bound exactly recovers the minimax bound while theirs exhibits a slight gap.

**Discussion on the optimality of $c_\phi$ dependency**    As discussed above, it is clear that the dependence on $c_\phi$ is tight for the standard cases of external, internal, and swap regret, since in these settings $c_\phi$ respectively equals 1, 1, and $d$, matching the known lower bounds. In fact, via a simple argument, one can establish a stronger lower bound showing that for any integer $k \in [d]$ and any algorithm, there exists a $d$-expert problem and a comparator mapping $\phi$ with $c_\phi \leq k + 1$, such that $\text{Reg}(\phi) = \Omega\left(\sqrt{c_\phi T \log c_\phi}\right)$. To see this, consider the following construction. Let $d - k$ experts be *dummy experts* that always incur the maximum loss of 1, while the remaining $k$ experts follow the swap-regret lower bound instance of Ito [2020], scaled by a factor of $1/2$. We define $\phi \in \Phi_b$ as follows. For each non-dummy expert, $\phi$ maps it optimally to another non-dummy expert (minimizing the total loss after swapping). For dummy experts, we distinguish two cases: if the algorithm selects dummy experts more than $\sqrt{kT \log k}$ times, we let $\phi$ map all dummy experts to a single fixed non-dummy expert

---

**Algorithm 1** MWU over $\Phi_b$ with prior $\pi$

---

**Input:** learning rate $\eta > 0$ and prior distribution $\pi$ defined in Definition 3.2. Initialize $q_1$ as $\pi$.

**for** $t = 1, 2 \ldots, T$ **do**

    Propose $\phi_t = \mathbb{E}_{\phi \sim q_t}[\phi] \in \mathcal{S}$ and receive loss matrix $p_t \ell_t^\top \in [0, 1]^{d \times d}$.

    Update $q_{t+1}$ such that $q_{t+1}(\phi) \propto q_t(\phi) \exp\left(-\eta \left\langle \phi, p_t \ell_t^\top \right\rangle\right)$.

---

(so that $d_\phi^{\text{unif}} \geq d - k$); otherwise, each dummy expert maps to itself (so that $d_\phi^{\text{self}} \geq d - k$). In both cases, we have $c_\phi \leq k + 1$.

In the first case, the regret satisfies $\text{Reg}(\phi) \geq \frac{1}{2}\sqrt{kT \log k}$, since whenever the algorithm chooses a dummy expert $i$, it incurs loss 1 while $\phi(e_i)$ incurs loss at most $1/2$. In the second case, $\text{Reg}(\phi)$ corresponds to the swap regret of the algorithm on a $k$-expert problem that lasts for at least $T - \sqrt{kT \log k}$ rounds. Because this instance follows the lower bound construction of Ito [2020], we again obtain $\text{Reg}(\phi) \geq \Omega(\sqrt{kT \log k})$. This shows that the dependence on $c_\phi$ in our upper bound is tight.

## 4 First Approach: Learning over Multiple Kernelized MWU's

In this section, we introduce our first approach to achieve Eq. (3). As mentioned, based on standard analysis (see e.g., [Freund and Schapire, 1999]), simply running MWU (Algorithm 1) with expert set $\Phi_b$, a fixed learning rate $\eta > 0$, and our prior distribution $\pi$ defined in Eq. (1) to get $q_t \in \Delta(\Phi_b)$ and outputting the stationary distribution of $\mathbb{E}_{\phi \sim q_t}[\phi]$ already gives $\text{Reg}(\phi) \leq \frac{\text{KL}(q, \pi)}{\eta} + \eta T$ for any $\phi \in \mathcal{S}$ and $q \in Q_\phi$ (recall $Q_\phi$ defined in Definition 2.1), which further implies $\text{Reg}(\phi) \leq \frac{\log(1/\pi(\phi))}{\eta} + \eta T$ for any $\phi \in \Phi_b$. With the "optimal tuning" of $\eta$, Eq. (3) would have been achieved. However, there is no such fixed "optimal tuning" since we require the bound to hold for all $\phi$ simultaneously, and different $\phi$ might lead to different optimal tuning. We will first address this issue using a simple idea, before addressing the other obvious issue that naively running MWU is computationally inefficient.

**Learning the learning rate via a meta MWU**  While there are many different ways to handle the aforementioned issue of parameter tuning (see e.g., Luo and Schapire [2015], Koolen and Van Erven [2015]), we resort to the most basic idea of learning the learning rate via another meta MWU, which is important for resolving the computational inefficiency later; see Algorithm 2 for the pseudocode. Specifically, the meta MWU learns over and combines decisions from a set of $2\lceil \log_2 d \rceil$ base learners, the $h$-th of which is an instance of MWU (Algorithm 1) with learning rate $\eta_h = \sqrt{2^h/T}$. This ensures that the optimal learning rate of interest always lies in $[\eta_h, 2\eta_h]$ for certain $h$. At each round $t$, the meta MWU maintains a distribution $w_t$ over all base learners. After receiving $\phi_t^h$, the expected transformation matrix from each base learner $\mathcal{B}_h$, the meta MWU computes the weighted average of them using $w_t$ and proposes $p_t$ as the stationary distribution of this weighted average.[4] Then, after receiving the loss vector $\ell_t$, the meta MWU constructs the loss $\ell_{t,h}^w \triangleq \left\langle \phi_t^h, p_t \ell_t^\top \right\rangle$ for each $\mathcal{B}_h$ and updates its weight $w_t$ via an exponential weight update. Finally, the meta MWU sends the loss matrix $p_t \ell_t^\top$ to each base learner $\mathcal{B}_h$. It is straightforward to prove the following result.

**Theorem 4.1.** *Algorithm 2 guarantees $\text{Reg}(\phi) = \mathcal{O}\left(\sqrt{T\text{KL}(q, \pi)} + \sqrt{T \log \log d}\right)$ for any $\phi \in \mathcal{S}$ and $q \in Q_\phi$. Consequently, it also guarantees Eq. (3) with $B = \sqrt{T \log \log d}$ and thus $\text{Reg}(\phi) = \mathcal{O}\left(\sqrt{(1 + c_\phi \log d)T} + \sqrt{T \log \log d}\right)$ for any $\phi \in \mathcal{S}$.*

The bound in terms of $\sqrt{T\text{KL}(q, \pi)}$ is stronger than what we need in Eq. (3), and using this stronger version in fact also allows us to additionally obtain the near-optimal $\varepsilon$-quantile regret bound of order $\mathcal{O}(\sqrt{T \log(1/\varepsilon)} + \sqrt{T \log \log d})$ when competing with the top $\varepsilon$-quantile of experts [Chaudhuri et al., 2009]; see Theorem B.3 for details.

---

[4]We remark that it is important to use the stationary distribution of the weighted average of $\phi_t^h$, but not the weighted average of the stationary distribution of $\phi_t^h$.

---

**Algorithm 2** Meta MWU Algorithm

---

**Initialization:** Set $\eta = \sqrt{\frac{\log\log d}{T}}$, $M = 2\lceil\log_2 d\rceil$, and $w_1 = \frac{1}{M}\cdot\mathbf{1} \in \Delta(M)$; initialize $M$ instances of Algorithm 1 (or Algorithm 3) $\{\mathcal{B}_h\}_{h=1}^M$ with the learning rate for $\mathcal{B}_h$ being $\eta_h = \sqrt{2^h/T}$.

**for** $t = 1, 2, \cdots, T$ **do**

> Receive $\phi_t^h = \mathbb{E}_{\phi \sim q_t^h}[\phi]$ from $\mathcal{B}_h$ for each $h \in [M]$ and compute $\phi_t = \sum_{h=1}^M w_{t,h}\phi_t^h$.
> Play the stationary distribution $p_t$ of $\phi_t$ (that is, $p_t = \phi_t(p_t)$) and receive loss $\ell_t$.
> Update $w_{t+1}$ such that $w_{t+1,h} \propto w_{t,h}\exp\left(-\eta\ell_{t,h}^w\right)$, where $\ell_{t,h}^w = \langle\phi_t^h, p_t\ell_t^\top\rangle$ for each $h \in [M]$.
> Send loss matrix $p_t\ell_t^\top$ to $\mathcal{B}_h$ for each $h \in [M]$.

---

---

**Algorithm 3** Kernelized MWU with non-uniform prior

---

**Input:** learning rate $\eta > 0$ and prior distribution $\pi$ (Definition 3.2); initialize $B_1 = \mathbf{1}\mathbf{1}^\top \in \mathbb{R}^{d\times d}$.

**for** $t = 1, 2, \cdots, T$ **do**

> Compute $\phi_t \in \mathcal{S}$ such that $(\phi_t)_{ij} = 1 - \frac{K(B_t, \mathbf{1}\mathbf{1}^\top - e_i e_j^\top)}{K(B_t, \mathbf{1}\mathbf{1}^\top)}$.
> Receive $p_t\ell_t^\top$ and update $B_{t+1} \in \mathbb{R}^{d\times d}$ such that $(B_{t+1})_{ij} = (B_t)_{ij}\cdot\exp(-\eta(p_t\ell_t^\top)_{ij})$.

---

**Efficient Implementation of Algorithm 1 via Kernelization**   To address the computational ineffi-ciency of Algorithm 1, we take inspiration from Farina et al. [2022b] that shows that Algorithm 1 with a uniform prior can be simulated efficiently as long as a certain kernel function can be evaluated efficiently, and extend their idea from uniform prior to non-uniform prior. Specifically, we propose the following prior-dependent kernel function.

**Definition 4.2** (kernel function). *Define kernel $K(B, A) = \sum_{\phi \in \Phi_b}\pi(\phi)\prod_{i,j\in[d]:\phi_{ij}=1}B_{ij}A_{ij}$ for any $B, A \in \mathbb{R}^{d\times d}$.*

We then show that this kernel function can be evaluated efficiently thanks to the structure of our prior $\pi$ and consequently the key output $\phi_t$ in Algorithm 1 (required for Algorithm 2) can also be computed efficiently via the Kernelized MWU shown in Algorithm 3.

**Theorem 4.3.** *The kernel function $K$ defined in Definition 4.2 can be evaluated in time $O(d^3)$. Moreover, the $\phi_t$ matrix computed by Algorithm 1 and Algorithm 3 are exactly the same.*

This theorem already shows that each iteration of Algorithm 3 can be implemented in time $\mathcal{O}(d^5)$ since it requires evaluating the kernel $2d^2$ times. However, by reusing some intermediate statistics that are common in these $2d^2$ kernel evaluations and the special structure of the stochastic matrices $\psi^1, \ldots, \psi^{d+1}$ defined in Eq. (2), we can further speed up the algorithm such that each iteration takes only $\mathcal{O}(d^2)$ time, making our algorithm as efficient as those by Blum and Mansour [2007], Lu et al. [2025]; see Appendix B.3.2 for details.

Combining Theorem 4.3 and Theorem 4.1, we have thus shown that Algorithm 2 is an efficient algorithm with regret $\text{Reg}(\phi) = \mathcal{O}(\sqrt{(1 + c_\phi\log d)T} + \sqrt{T\log\log d})$ for all $\phi \in \mathcal{S}$ simultaneously.

## 5   Second Approach: Learning over Multiple BM-Reductions

In this section, we introduce our second approach to achieve Eq. (3) using a prior-aware variant of the BM-reduction [Blum and Mansour, 2007]. As a reminder, BM-reduction reduces swap regret minimization to $d$ external regret minimization problems, each with a different scaled loss vector in each round, and achieves $\text{Reg}(\phi) \le \frac{d\log d}{\eta} + \eta T$ when each base external regret minimization algorithm is MWU (over $[d]$) with learning rate $\eta$. Given a prior $\pi \in \Delta(\Phi_b)$, it is natural to ask whether a variant of BM-reduction can achieve $\text{Reg}(\phi) \le \frac{\log(1/\pi(\phi))}{\eta} + \eta T$, replacing $d\log d$ with $\log(1/\pi(\phi))$. We first show that this is indeed possible, but only when $\pi$ is a $\psi$-induced distribution for some $\psi \in \mathcal{S}$ (Definition 3.1), and the only modification needed is to let the $i$-th MWU subroutine use the prior $\psi_{i:} \in \Delta(d)$. See Theorem C.1 in Appendix C for details.

Given that our prior of interest is a mixture of $d+1$ distributions induced by $\psi^1, \ldots, \psi^{d+1}$ (Definition 3.2) and also the same issue that a fixed learning rate $\eta$ cannot be adaptive to different comparator $\phi$, we propose a natural meta-base framework that is very similar to Algorithm 2 and learns over both different $\psi^k$ and different learning rates. Specifically, we maintain $(d+1)M$ (where $M$ is again $2\lceil \log_2 d \rceil$) base-learners $\mathcal{B}_{k,h}$, indexed by $k \in [d+1]$ and $h \in [M]$. Each base-learner $\mathcal{B}_{k,h}$ is an instance of the prior-aware BM-reduction Algorithm 8 with prior $\psi^k$ and learning rate $\sqrt{2^h/T}$. With this set of base learners, the rest of the algorithm is exactly the same as Algorithm 2, and we thus defer all details to Algorithm 7 in the appendix. The only crucial point (similar to Footnote 4) is that, even though the standard BM reduction directly outputs the stationary distribution of a stochastic matrix, it is important here that we first take a convex combination of these stochastic matrices and then compute its stationary distribution, instead of using the convex combination of stationary distributions.

The following theorem shows that Algorithm 7 satisfies Eq. (3) with $B = \sqrt{T \log d}$.

**Theorem 5.1.** *Algorithm 7 satisfies Eq. (3) with $B = \sqrt{T \log d}$. Consequently, it guarantees* $\mathrm{Reg}(\phi) = \mathcal{O}\left( \sqrt{(1 + c_\phi \log d)T} + \sqrt{T \log d} \right)$ *for any $\phi \in \mathcal{S}$.*

Even though the guarantee of this second approach is slightly worse than that of Algorithm 2 (but still better than Lu et al. [2025]), in the next section, we show that its particular structure is crucial in extending our results to games.

## 6 Applications to Games

In this section, we discuss how to extend Algorithm 7 to achieve accelerated and adaptive $\Phi$-equilibrium convergence in $N$-player general-sum normal-form games. We first introduce necessary background on the connection between online learning and games. Consider an $N$-player general-sum normal-form game, where each player $n \in [N]$ has a finite set of actions $[d]$.[5] For a given joint action profile $\mathbf{a} = (a_1, \ldots, a_N) \in [d]^N \triangleq \mathcal{A}$, the loss received by player $n$ is given by some loss function $\ell^{(n)} : \mathcal{A} \to [0,1]$. For notational convenience, denote $\mathbf{a}^{(-n)} = (a_1, \ldots, a_{n-1}, a_{n+1}, \ldots, a_N)$. Given $\Phi = \times_{n=1}^N \Phi_n$ where each $\Phi_n \subseteq \mathcal{S}$ is a set of action transformations for player $n$, the corresponding (approximate) $\Phi$-equilibrium is defined as follows.

**Definition 6.1** ($\varepsilon$-approximate $\Phi$-equilibrium). *We call a distribution $\boldsymbol{p} \in \Delta(\mathcal{A})$ over all joint action profiles an $\varepsilon$-approximate $\Phi$-equilibrium if for all players $n \in [N]$ and all $\phi \in \Phi_n$, $\mathbb{E}_{\mathbf{a} \sim \boldsymbol{p}}[\ell^{(n)}(\mathbf{a})] \le \mathbb{E}_{\mathbf{a} \sim \boldsymbol{p}}[\ell^{(n)}(\phi(a^{(n)}), \mathbf{a}^{(-n)})] + \varepsilon$. When $\varepsilon = 0$, we call $\boldsymbol{p}$ a $\Phi$-equilibrium.*

When $\Phi_n = \Phi_{\mathsf{Ext}}$ for all $n$, $\Phi$-equilibrium reduces to *Coarse Correlated Equilibrium* (CCE), and when $\Phi_n = \Phi_b$ for all $n$, $\Phi$-equilibrium reduces to *Correlated Equilibrium* (CE).

Approximate $\Phi$-equilibrium can be found via the following uncoupled no-regret learning dynamic. At each round $t \in [T]$, each player $n$ proposes $p_t^{(n)} \in \Delta(d)$, forming an uncorrelated distribution $\boldsymbol{p}_t = (p_t^{(1)}, \ldots, p_t^{(N)})$, and receives a loss vector $\ell_t^{(n)} \in [0,1]^d$ as the feedback where $\ell_{t,a}^{(n)} \triangleq \mathbb{E}_{\mathbf{a} \sim \boldsymbol{p}_t}[\ell^{(n)}(a, \mathbf{a}^{(-n)})]$, for any $a \in [d]$. The $\Phi_n$-regret for player $n$ is then defined as $\mathrm{Reg}_n \triangleq \max_{\phi \in \Phi_n} \mathrm{Reg}_n(\phi) = \max_{\phi \in \Phi_n} \sum_{t=1}^T \langle p_t^{(n)} - \phi(p_t^{(n)}), \ell_t^{(n)} \rangle$, and we denote the special case of external regret for $\Phi_n = \Phi_{\mathsf{Ext}}$ as $\mathrm{Reg}_n^{\mathsf{Ext}}$. The following proposition from Greenwald and Jafari [2003] builds the connection between no-$\Phi$-regret learning and convergence to $\Phi$-equilibrium.

**Proposition 6.2** ([Greenwald and Jafari, 2003]). *The empirical distribution of joint strategy profiles, that is, uniform over $\boldsymbol{p}_1, \ldots, \boldsymbol{p}_T$, is a $\frac{\max_{n \in [N]}\{\mathrm{Reg}_n\}}{T}$-approximate $\Phi$-equilibrium.*

While one can apply our Algorithm 2 or Algorithm 7 directly for each player to obtain $1/\sqrt{T}$ convergence rate that is adaptive to the complexity of $\Phi$, we are interested in achieving accelerated $\widetilde{\mathcal{O}}(1/T)$ convergence rate that has been shown possible in recent years for canonical $\Phi$. For example, for CCE, Daskalakis et al. [2021], Farina et al. [2022a], Soleymani et al. [2025] show the following $\mathrm{polylog}(T)$ bound on $\mathrm{Reg}_n^{\mathsf{Ext}}$ respectively: $\mathcal{O}(N \log d \log^4 T), \mathcal{O}(Nd \log T), \mathcal{O}(N \log^2 d \log T)$;

---

[5]For notational conciseness, we assume that the action set size is the same for all players, but our analysis can be directly extended to games with different action set sizes.

---
**Algorithm 4** Meta Algorithm for Accelerated and Adaptive Convergence in Games
---
**Input:** learning rate $\eta_m > 0$, correction scale $\lambda > 0$

1    **Initialize:** $d + 2$ base learners $\mathcal{B}_1, \ldots, \mathcal{B}_{d+2}$, all with learning rate $\eta = \frac{1}{16N}$. For $k < d + 2$, $\mathcal{B}_k$ is an instance of Algorithm 8 with prior $\psi^k$ and SubAlg being OMWU (Algorithm 9); $\mathcal{B}_{d+2}$ is an instance of Algorithm 9 (with uniform prior); set $\widehat{w}_1 = [\frac{1}{2d}, \ldots, \frac{1}{2d}, \frac{1}{4}, \frac{1}{4}] \in \Delta(d + 2)$.

     **for** $t = 1, 2 \ldots, T$ **do**

2      Receive $\phi_t^k \in \mathcal{S}$ from base learner $\mathcal{B}_k$ for each $k \in [d + 2]$.

3      Compute $c_t \in \mathbb{R}^{d+2}$ where $c_{t,k} = \lambda \|\widetilde{p}_{t-1}^k - \widetilde{p}_{t-2}^k\|_1^2 \cdot \mathbb{1}\{t \geq 3\}$ and $\widetilde{p}_t^k = \phi_t^k(p_t)$.

4      Compute $m_t^w \in \mathbb{R}^{d+2}$ where $m_{t,k}^w = \langle \phi_t^k, p_{t-1}\ell_{t-1}^\top \rangle \cdot \mathbb{1}\{t \geq 2\}$ for $k \in [d + 2]$.

5      Compute $w_t$ such that $w_{t,k} \propto \widehat{w}_{t,k} \exp(-\eta_m(m_{t,k}^w + c_{t,k}))$.

6      Compute $\phi_t = \sum_{k=1}^{d+2} w_{t,k}\phi_t^k$ and play stationary distribution $p_t$ satisfying $p_t = \phi_t(p_t)$.

7      Receive $\ell_t$ and compute $\ell_t^w \in \mathbb{R}^{d+2}$ where $\ell_{t,k}^w = \langle \phi_t^k, p_t\ell_t^\top \rangle$ for $k \in [d + 2]$.

8      Update $\widehat{w}_{t+1}$ such that $\widehat{w}_{t+1,k} \propto \widehat{w}_{t,k} \exp(-\eta_m(\ell_{t,k}^w + c_{t,k}))$.

9      Send $p_t\ell_t^\top$ to $\mathcal{B}_k$ for $k \in [d + 1]$ and send $\ell_t$ to $\mathcal{B}_{d+2}$.
---

for CE, Anagnostides et al. [2022a,b] show the following bound on $\mathrm{Reg}_n$: $\mathcal{O}(Nd \log d \log^4 T)$ and $\mathcal{O}(Nd^{2.5} \log T)$. Our goal is to achieve similar fast rates while at the same time being adaptive to the complexity of $\Phi$, and we successfully achieve so, albeit only for the following class of games.

**Definition 6.3** (Nonnegative-social-external-regret games). *We call a game a* nonnegative-social-external-regret game *if* $\sum_{n=1}^N \mathrm{Reg}_n^{\mathsf{Ext}} \geq 0$ *always holds.*

This class was explicitly considered in Anagnostides et al. [2022c] and contains a broad family of well-studied games, including constant-sum polymatrix games, polymatrix strategically zero-sum games, and quasiconvex-quasiconcave games. Therefore, we believe that our results are still very general and non-trivial. We are unable to deal with general games using ideas from aforementioned recent work due to the two-layer nature of our algorithms. In fact, even when considering only this subclass of games, it is unclear to us how to make our first approach discussed in Section 4 or the algorithm of Lu et al. [2025] work, and we have to resort to extending our Algorithm 7. In the following, we discuss how we design our algorithm (shown in Algorithm 4) based on similar ideas of Algorithm 7 and what extra ingredients are needed.

**Base learners**    Compared to Algorithm 7, there are several differences in the base learner design. First, while we still maintain a base learner $\mathcal{B}_k$ (Algorithm 8) for each prior $\psi^k$, we do not need to maintain different copies of it to account for different learning rates, since in the end we will use a fixed constant learning rate, similar to prior work on accelerated convergence. Second, inspired by a long line of work showing that optimism accelerates convergence, for each $\mathcal{B}^k$, we replace its subroutines from MWU to Optimistic MWU (OMWU) [Rakhlin and Sridharan, 2013, Syrgkanis et al., 2015] (Algorithm 9). Finally, besides these $d + 1$ base learners, we additionally include a base learner $\mathcal{B}_{d+2}$, an instance of OMWU (Algorithm 9) with a uniform prior, to explicitly minimize external regret. This last modification is in a way most crucial to our analysis, since it allows us to utilize the nonnegative-social-external-regret property and show that the path-length of the entire learning dynamic is $T$-independent and of order $\mathcal{O}(N \log d)$ only; see Appendix D.3 for details.

**Meta learner**    In addition, there are also several modifications to the meta learner compared to Algorithm 7. First, similar to the base learners, instead of using MWU, we apply OMWU to compute $w_t$ and the auxiliary $\widehat{w}_t$ (Line 5 and Line 8 of Algorithm 4). Importantly, the update of $w_t$ uses a "predictive loss vector" $m_t^w$ such that $m_{t,k}^w = \langle \phi_t^k, p_{t-1}\ell_{t-1}^\top \rangle$ (Line 4). The fact that $m_t^w$ is not simply the previous loss vector $\ell_{t-1}^w$, a canonical setup for OMWU, is important for the analysis, as already shown in Zhang et al. [2022] under a different context. Second, also inspired by Zhang et al. [2022], Zhao et al. [2024], in order to aggregate the guarantee for all base learners, in both the update of $w_t$ and $\widehat{w}_t$, we propose to add a stability correction term $c_t$ (Line 3 of Algorithm 4), which guides the meta algorithm to bias toward the more stable base learners, hence also stabilizing the final decision. While the idea is similar, the specific value of $c_{t,k}$ is tailored to our analysis and takes into account not only the stability of $\phi_t^k$ from the base learner $\mathcal{B}_k$ but also the stability of the stationary distribution $p_t$. Our main result is as follows.

**Theorem 6.4.** *For an $N$-player normal-form general-sum game satisfying Definition 6.3, if each player $n \in [N]$ runs Algorithm 4 with $\eta_m = \frac{1}{64N}$ and $\lambda = N$, then we have $\mathrm{Reg}_n(\phi) = \mathcal{O}(c_\phi N \log d + N^2 \log d)$ and $\mathrm{Reg}_n^{\mathsf{Ext}} = \mathcal{O}(N \log d)$ for all $n \in [N]$. Consequently, the uniform distribution over their joint strategy profiles is an $\mathcal{O}\left(\frac{N \log d}{T}\right)$-approximate CCE and also an $\mathcal{O}\left(\frac{\max_{n \in [N], \phi \in \Phi_n} c_\phi N \log d + N^2 \log d}{T}\right)$-approximate $\Phi$-equilibrium, simultaneously for all $\Phi \subseteq \mathcal{S}^N$.*

To our knowledge, our result achieves the first adaptive and accelerated $\Phi$-equilibrium guarantee. For the special case of CCE, the rate $\mathcal{O}(\frac{N \log d}{T})$ matches that of OMWU (for nonnegative-social-external-regret games), and for CE, it is unclear at all what better results one can obtain for nonnegative-social-external-regret games than those rates from [Anagnostides et al., 2022a,b] for general games. If we compare their bounds to ours, since $\max_{n \in [N], \phi \in \Phi_n} c_\phi = d$ in this case, we improve over Anagnostides et al. [2022b] on the $d$-dependence and remove any $\mathrm{polylog}(T)$ dependence compared to Anagnostides et al. [2022a,b]. One disadvantage of our results is the additive term of $N^2 \log d$ for $\Phi$-equilibrium other than CCE. Removing this term is an interesting future direction.

# 7  Conclusion and Future Directions

In this work, we significantly improve over a recent work by Lu et al. [2025] regarding comparator adaptive $\Phi$-regret, by developing simpler algorithms, better bounds, and broader applications to games. The most interesting future direction is to improve our results for games, especially to remove the requirement on nonnegative social external regret. The idea of high-order stability from Daskalakis et al. [2021], Anagnostides et al. [2022a] might be useful, but appropriately combining this idea with our approaches requires further investigation. For the expert problem, it is also interesting to derive comparator-adaptive $\Phi$-regret with respect to other complexity measure of the comparator.

**Acknowledgement**  HL thanks Shinji Ito for initial discussion on this topic. He is supported by NSF award IIS-1943607.

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

# A  Omitted Details in Section 2 and Section 3

As mentioned, Lu et al. [2025] define $c_\phi$ as $\min\{d - d_\phi^{\text{self}}, d - d_\phi^{\text{unif}} + 1\}$ for all $\phi \in \mathcal{S}$, while our definition for $\phi \in \mathcal{S} \setminus \Phi_b$ is different. The following shows that ours is strictly better.

**Proposition A.1.** *For any $\phi \in \mathcal{S}$, we have $c_\phi \leq \min\{d - d_\phi^{\text{self}}, d - d_\phi^{\text{unif}} + 1\}$. Moreover, there exists $\phi \in \mathcal{S}$ such that $c_\phi = \mathcal{O}(1)$ and $\min\{d - d_\phi^{\text{self}}, d - d_\phi^{\text{unif}} + 1\} = \Omega(d)$.*

*Proof of Proposition A.1.* Since

$$
\begin{aligned}
c_\phi &= \min_{q \in Q_\phi} \mathbb{E}_{\phi' \sim q}[c_{\phi'}] = \min_{q \in Q_\phi} \mathbb{E}_{\phi' \sim q}[\min\{d - d_{\phi'}^{\text{self}}, d - d_{\phi'}^{\text{unif}} + 1\}] \\
&\leq \min_{q \in Q_\phi} \min\{\mathbb{E}_{\phi' \sim q}[d - d_{\phi'}^{\text{self}}], \mathbb{E}_{\phi' \sim q}[d - d_{\phi'}^{\text{unif}} + 1]\} \\
&= \min\{\min_{q \in Q_\phi} \mathbb{E}_{\phi' \sim q}[d - d_{\phi'}^{\text{self}}], \min_{q \in Q_\phi} \mathbb{E}_{\phi' \sim q}[d - d_{\phi'}^{\text{unif}} + 1]\} \\
&= \min\{d - \operatorname{trace}(\phi), d - \max_{q \in Q_\phi} \mathbb{E}_{\phi' \sim q}[d_{\phi'}^{\text{unif}}] + 1\},
\end{aligned}
$$

it suffices to prove $d_\phi^{\text{self}} \leq \operatorname{trace}(\phi)$ and $d_\phi^{\text{unif}} \leq \max_{q \in Q_\phi} \mathbb{E}_{\phi' \sim q}[d_{\phi'}^{\text{unif}}]$ separately. First, by the definition of $d_\phi^{\text{self}}$, it directly follows that $d_\phi^{\text{self}} \leq \operatorname{trace}(\phi)$.

Second, we construct a distribution $p \in \Delta(\Phi_b)$ such that $\phi = \sum_{\phi' \in \Phi_b} p(\phi')\phi'$ and show that $\sum_{\phi' \in \Phi_b} p(\phi')d_{\phi'}^{\text{unif}} \geq d_\phi^{\text{unif}}$, which in turn implies that $\max_{q \in Q_\phi} \mathbb{E}_{\phi' \sim q}[d_{\phi'}^{\text{unif}}] \geq d_\phi^{\text{unif}}$. Suppose that the most frequent element in $\{\phi(e_1), \cdots, \phi(e_d)\}$ is $q \in \Delta(d)$ and $\phi(e_j) = q$ for all $j \in \mathcal{A} \subseteq [d]$ with $|\mathcal{A}| = d_\phi^{\text{unif}}$. Then, we can write $\phi$ as $\phi = \sum_{i=1}^d q_i \phi_i$, where $\phi_i(e_j) = e_i$ for all $j \in \mathcal{A}$ and $\phi_i(e_j) = \phi(e_j)$ for all $j \notin \mathcal{A}$. This guarantees that $d_{\phi_i}^{\text{unif}} \geq d_\phi^{\text{unif}}$. Furthermore, let $\phi_i = \sum_{\phi'_i \in \Phi_b} p_i(\phi'_i) \cdot \phi'_i$ be any convex decomposition of $\phi_i$, and note that for any $\phi'_i$ in the support of $p_i$, we must have $\phi'_i(e_j) = e_i$ for all $j \in \mathcal{A}$, meaning that $d_{\phi'_i}^{\text{unif}} \geq d_{\phi_i}^{\text{unif}}$. Now we have constructed a convex combination for $\phi$:

$$
\phi = \sum_{i=1}^d q_i \phi_i = \sum_{i=1}^d \sum_{\phi'_i \in \Phi_b} q_i \cdot p_i(\phi'_i) \cdot \phi'_i
$$

and consequently,

$$
d_\phi^{\text{unif}} \leq \sum_{i=1}^d q_i \cdot d_{\phi_i}^{\text{unif}} \leq \sum_{i=1}^d \sum_{\phi'_i \in \Phi_b} q_i \cdot p_i(\phi'_i) \cdot d_{\phi'_i}^{\text{unif}} \leq \max_{q \in Q_\phi} \mathbb{E}_{\phi' \sim q}[d_{\phi'}^{\text{unif}}].
$$

This proves that $d_\phi^{\text{unif}} \leq \max_{q \in Q_\phi} \mathbb{E}_{\phi' \sim q}[d_{\phi'}^{\text{unif}}]$. Combining with $d_\phi^{\text{self}} \leq \operatorname{trace}(\phi)$, we have shown $c_\phi \leq \min\{d - d_\phi^{\text{self}}, d - d_\phi^{\text{unif}} + 1\}$.

Moreover, the complexity measures $c_\phi$ and $\min\{d - d_\phi^{\text{self}}, d - d_\phi^{\text{unif}} + 1\}$ can differ significantly in some cases. For example, when $\phi_1$ is a row-stochastic matrix with all diagonal entries equal to $1 - \varepsilon$ for small $\varepsilon > 0$ (implying each row's off-diagonal entries sum to $\varepsilon$), $\operatorname{trace}(\phi_1)$ is $(1 - \varepsilon)d$ while $d_{\phi_1}^{\text{self}}$ is 0. Similarly, when $\phi_2 = (1 - \varepsilon) \cdot \mathbf{1}_d \cdot e_1^\top + \varepsilon \mathbf{I}$, it can be verified that $d_{\phi_2}^{\text{unif}} = 1$ and $\max_{q \in Q_{\phi_2}} \mathbb{E}_{\phi' \sim q}[d_{\phi'}^{\text{unif}}] = (1 - \varepsilon)d + \varepsilon = d - (d - 1)\varepsilon$. With $\varepsilon < \frac{1}{d}$, we have $c_\phi = \mathcal{O}(1)$ and $\min\{d - d_\phi^{\text{self}}, d - d_\phi^{\text{unif}} + 1\} = \Omega(d)$ for $\phi = \phi_1, \phi_2$.  $\square$

Next, we prove Theorem 3.3.

*Proof of Theorem 3.3.* First, for $\pi_{\psi^{d+1}}$, we have for any $\phi \in \Phi_b$,

$$
\begin{aligned}
\pi_{\psi^{d+1}}(\phi) &= \left(1 - \frac{1}{d}\right)^{d_\phi^{\text{self}}} \cdot \left(\frac{1}{d(d-1)}\right)^{d - d_\phi^{\text{self}}} \\
&\geq \left(1 - \frac{1}{d}\right)^{d_\phi^{\text{self}}} \cdot \frac{1}{d^{2(d - d_\phi^{\text{self}})}},
\end{aligned}
$$

which leads to

$$\log \frac{1}{\pi_{\psi^{d+1}}(\phi)} \leq 2(d - d_\phi^{\text{self}}) \log d + 1, \tag{4}$$

since $-d_\phi^{\text{self}} \log \left(1 - \frac{1}{d}\right) \leq -d \log \left(1 - \frac{1}{d}\right) \leq 1$ for $d \geq 2$. Then, for $\phi \in \Phi_b$, assume that the most frequent element in the set $\{\phi(e_1), \ldots, \phi(e_d)\}$ is $e_r$. It holds that

$$\pi_{\psi^r}(\phi) = \left(1 - \frac{1}{d}\right)^{d_\phi^{\text{unif}}} \cdot \left(\frac{1}{d(d-1)}\right)^{d - d_\phi^{\text{unif}}}$$

$$\geq \left(1 - \frac{1}{d}\right)^{d_\phi^{\text{unif}}} \cdot \frac{1}{d^{2(d - d_\phi^{\text{unif}})}},$$

which leads to

$$\log \frac{1}{\pi_{\psi^r}(\phi)} \leq 2(d - d_\phi^{\text{unif}}) \log d + 1, \tag{5}$$

since $-d_\phi^{\text{unif}} \log \left(1 - \frac{1}{d}\right) \leq -d \log \left(1 - \frac{1}{d}\right) \leq 1$ for $d \geq 2$. Using the definition of $\pi$ in Definition 3.2 and combining Eq. (4) and Eq. (5), we have

$$\log \frac{1}{\pi(\phi)} \leq \min \left\{ \log \frac{1}{\frac{1}{2} \cdot \pi_{\psi^{d+1}}(\phi)}, \log \frac{1}{\frac{1}{2d} \cdot \pi_{\psi^r}(\phi)} \right\}$$

$$\leq \min \left\{ 2(d - d_\phi^{\text{unif}}) \log d + 1 + \log 2, 2(d - d_\phi^{\text{unif}}) \log d + 1 + \log(2d) \right\}$$

$$\leq 2 \min\{d - d_\phi^{\text{self}}, d - d_\phi^{\text{unif}} + 1\} \cdot \log d + 2. \tag{6}$$

This completes the proof of the first statement that $\log \left(\frac{1}{\pi(\phi)}\right) \leq 2 + 2c_\phi \log d$.

Next, we prove that $\text{Reg}(\phi) = \mathcal{O}\left(\sqrt{(1 + c_\phi \log d)T} + B\right)$ for all $\phi \in \mathcal{S}$ when the condition Eq. (3) holds. Fix a row-stochastic matrix $\phi \in \mathcal{S}$ and let $q \in Q_\phi$ be such that $c_\phi = \mathbb{E}_{\phi' \sim q}[c_{\phi'}]$. By linearity of $\text{Reg}(\phi)$ in $\phi$, we have $\text{Reg}(\phi) = \mathbb{E}_{\phi' \sim q}[\text{Reg}(\phi')]$, and thus

$$\text{Reg}(\phi) \leq \mathbb{E}_{\phi' \sim q}\left[\sqrt{T \log \left(\frac{1}{\pi(\phi)}\right)} + B\right] \qquad \text{(by Eq. (3))}$$

$$\leq \mathbb{E}_{\phi' \sim q}\left[\sqrt{T\left(2c_{\phi'} \log d + 2\right)} + B\right] \qquad \text{(by Eq. (6))}$$

$$\leq \sqrt{T\left(2 \cdot \mathbb{E}_{\phi' \sim q}[c_{\phi'}] \log d + 2\right)} + B \qquad \text{(by Jensen's inequality)}$$

$$= \mathcal{O}\left(\sqrt{(1 + c_\phi \log d)T} + B\right).$$

This completes the proof. $\qquad \square$

# B  Omitted Details in Section 4

In this section, we show the omitted proofs in Section 4.

## B.1  Proof of Theorem 4.1

First, we provide the general form of vanilla MWU for an arbitrary and finite action space $\mathcal{A}$ in Algorithm 5. The following result is well-known for MWU, and we provide a proof for completeness.

**Lemma B.1.** *Algorithm 5 ensures that for any comparator $q \in \Delta(\mathcal{A})$, we have*

$$\sum_{t=1}^{T} \langle x_t - q, \ell_t \rangle \leq \frac{\text{KL}\left(q, x_1\right)}{\eta} + \eta \sum_{t=1}^{T} \|\ell_t\|_\infty^2.$$

---
**Algorithm 5** MWU
---
**Input:** learning rate $\eta > 0$; finite action space $\mathcal{A}$; prior distribution $x_1 \in \Delta(\mathcal{A})$.
**for** $t = 1, 2 \ldots, T$ **do**
  Play $x_t$ and receive loss $\ell_t \in [0, 1]^{|\mathcal{A}|}$.
  Update $x_{t+1}$ such that $x_{t+1,i} \propto x_{t,i} \exp\left(-\eta \ell_{t,i}\right)$ for all $i \in \mathcal{A}$.
---

*Proof of Lemma B.1.* We aim to show that for all $t \in [T]$,

$$\langle x_t - q, \ell_t \rangle = \frac{1}{\eta} \left( \text{KL}(q, x_t) - \text{KL}(q, x_{t+1}) + \text{KL}(x_t, x_{t+1}) \right). \tag{7}$$

Note that the update rule of MWU implies that $x_{t+1,i} = \frac{x_{t,i} \exp(-\eta \ell_{t,i})}{\sum_{j \in \mathcal{A}} x_{t,j} \exp(-\eta \ell_{t,j})}$ with prior distribution $x_1$. Direct calculation shows that

$$\frac{1}{\eta} \left( \text{KL}(q, x_t) - \text{KL}(q, x_{t+1}) + \text{KL}(x_t, x_{t+1}) \right)$$

$$= \frac{1}{\eta} \sum_{i \in \mathcal{A}} (x_{t,i} - q_i) \cdot \log \frac{x_{t,i}}{x_{t+1,i}}$$

$$= \sum_{i \in \mathcal{A}} (x_{t,i} - q_i) \left( \ell_{t,i} + \frac{1}{\eta} \log \left( \sum_{j \in \mathcal{A}} x_{t,j} \exp(-\eta \ell_{t,j}) \right) \right)$$

$$= \langle x_t - q, \ell_t \rangle.$$

Summing Eq. (7) for all $t \in [T]$, we obtain that

$$\sum_{t=1}^{T} \langle x_t - q, \ell_t \rangle = \frac{1}{\eta} \left( \text{KL}(q, x_1) - \text{KL}(q, x_{T+1}) \right) + \frac{1}{\eta} \sum_{t=1}^{T} \text{KL}(x_t, x_{t+1}). \tag{8}$$

Next, we bound $\text{KL}(x_t, x_{t+1})$ as shown below.

$$\text{KL}(x_t, x_{t+1}) = \sum_{i \in \mathcal{A}} x_{t,i} \log \frac{x_{t,i}}{x_{t+1,i}}$$

$$= \sum_{i \in \mathcal{A}} \eta x_{t,i} \ell_{t,i} + x_{t,i} \log \left( \sum_{j \in \mathcal{A}} x_{t,j} \exp(-\eta \ell_{t,j}) \right)$$

$$\leq \sum_{i \in \mathcal{A}} \eta x_{t,i} \ell_{t,i} + x_{t,i} \log \left( \sum_{j \in \mathcal{A}} x_{t,j} \left( 1 - \eta \ell_{t,j} + \eta^2 \ell_{t,j}^2 \right) \right)$$
$$\text{(since } \exp(-x) \leq 1 - x + x^2 \text{ for } x \geq -1)$$

$$= \sum_{i \in \mathcal{A}} \eta x_{t,i} \ell_{t,i} + x_{t,i} \log \left( 1 - \eta \sum_{j \in \mathcal{A}} x_{t,j} \ell_{t,j} + \eta^2 \sum_{j \in \mathcal{A}} x_{t,j} \ell_{t,j}^2 \right)$$

$$\leq \eta \sum_{i \in \mathcal{A}} x_{t,i} \ell_{t,i} - \eta \sum_{j \in \mathcal{A}} x_{t,j} \ell_{t,j} + \eta^2 \sum_{j \in \mathcal{A}} x_{t,j} \ell_{t,j}^2 \quad \text{(since } \log(1 + x) \leq x \text{ for all } x)$$

$$\leq \eta^2 \|\ell_t\|_\infty^2.$$

Substituting this in Eq. (8) and using the fact that KL divergence is always non-negative, we get,

$$\sum_{t=1}^{T} \langle x_t - q, \ell_t \rangle \leq \frac{\text{KL}(q, x_1)}{\eta} + \eta \sum_{t=1}^{T} \|\ell_t\|_\infty^2.$$

$\square$

The following lemma proves the statement in Section 4 that the optimal learning rate of interest always lies in $[\eta_h, 2\eta_h]$ for certain $h \in [M]$, where $M = 2\lceil \log_2 d \rceil$.

**Lemma B.2.** *For any $\phi \in \mathcal{S}$ and $q \in Q_\phi$, there exists $h \in [M]$, such that $\eta_h \leq \max\left\{\sqrt{\frac{\mathrm{KL}(q,\pi)}{T}}, \sqrt{\frac{2}{T}}\right\} \leq 2\eta_h$.*

*Proof of Lemma B.2.* For any $\phi \in \mathcal{S}$ and $q \in Q_\phi$, we have $\mathrm{KL}(q,\pi) = \sum_{\phi' \in \Phi_b} q(\phi') \log \frac{q(\phi')}{\pi(\phi')} \leq \sum_{\phi' \in \Phi_b} q(\phi') \log \frac{1}{\pi(\phi')} \leq \log \frac{1}{\min_{\phi' \in \Phi_b} \pi(\phi')} \leq 2d \log d + 1$, where the last inequality follows from the fact that $\min_{\phi' \in \Phi_b} \pi(\phi') \geq \min_{\phi' \in \Phi_b} \frac{1}{2} \pi_{\psi^{d+1}}(\phi') \geq \frac{1}{2} \cdot \left(\frac{1}{d(d-1)}\right)^d$. Therefore, we have

$$\min_{h \in [M]} 2^h = 2 \leq \max\{\mathrm{KL}(q,\pi), 2\} \leq 2d \log d + 1 \leq d^2 = 2^{2\log_2 d} \leq \max_{h \in [M]} 2^h,$$

and thus, there exists an $h \in [M]$, such that $\eta_h \leq \max\left\{\sqrt{\frac{\mathrm{KL}(q,\pi)}{T}}, \sqrt{\frac{2}{T}}\right\} \leq 2\eta_h$. $\qquad\square$

Now, we provide the proof of the adaptive $\Phi$-regret bound attained by the Meta MWU algorithm (Algorithm 2) in Section 4.

*Proof of Theorem 4.1.* Since $p_t$ is a stationary distribution of $\phi_t$, we have $\mathrm{Reg}(\phi) = \sum_{t=1}^{T} \langle p_t - \phi(p_t), \ell_t \rangle = \sum_{t=1}^{T} \langle \phi_t(p_t) - \phi(p_t), \ell_t \rangle = \sum_{t=1}^{T} \langle \phi_t - \phi, p_t \ell_t^\top \rangle$. Further using the definition of $\phi_t$ from Algorithm 2, we can express $\mathrm{Reg}(\phi)$ as the sum of the base and meta algorithms' regret as follows:

$$\mathrm{Reg}(\phi) = \sum_{t=1}^{T} \left\langle \sum_{h=1}^{M} w_{t,h} \phi_t^h - \phi, p_t \ell_t^\top \right\rangle$$
$$= \sum_{t=1}^{T} \langle w_t - e_{h^*}, \ell_t^w \rangle + \sum_{t=1}^{T} \left\langle \phi_t^{h^*} - \phi, p_t \ell_t^\top \right\rangle,$$

where $\ell_{t,h}^w = p_t^\top \phi_t^h \ell_t \in [0,1]$ and $h^* \in [M]$ is the index of an arbitrary base learner to be specified.

Regret of the meta MWU algorithm is bounded as $\sum_{t=1}^{T} \langle w_t - e_{h^*}, \ell_t^w \rangle \leq \mathcal{O}\left(\sqrt{T \log \log d}\right)$ from Lemma B.1 since the prior is the uniform distribution over the $2\lceil \log_2 d \rceil$ base algorithms.

Regret of the base MWU algorithm $\mathcal{B}_{h^*}$ is $\sum_{t=1}^{T} \left\langle \phi_t^{h^*} - \phi, p_t \ell_t^\top \right\rangle \leq \frac{\mathrm{KL}(q,\pi)}{\eta_{h^*}} + \eta_{h^*} T$ for any $q \in Q_\phi$ based on Lemma B.1 and Algorithm 1. Choosing $h^*$ according to Lemma B.2, we get $\sum_{t=1}^{T} \left\langle \phi_t^{h^*} - \phi, p_t \ell_t^\top \right\rangle \leq 3\sqrt{T \mathrm{KL}(q,\pi)} + 2\sqrt{2T}$. Thus, Algorithm 2 achieves $\mathrm{Reg}(\phi) = \mathcal{O}\left(\sqrt{T \mathrm{KL}(q,\pi)} + \sqrt{T \log \log d}\right)$ for all $\phi \in \mathcal{S}$ and $q \in Q_\phi$.

By selecting $q$ to be a one-hot vector that puts all weights on $\phi$, we obtain Eq. (3) with $B = \mathcal{O}\left(\sqrt{T \log \log d}\right)$ for any $\phi \in \Phi_b$. Combining with Theorem 3.3, we prove the desired bound of $\mathrm{Reg}(\phi) = \mathcal{O}\left(\sqrt{(1 + c_\phi \log d)T} + \sqrt{T \log \log d}\right)$ for all $\phi \in \mathcal{S}$. $\qquad\square$

## B.2 Quantile Regret

In this section, we show the near-optimal $\varepsilon$-quantile regret bound promised by Algorithm 2. The $\varepsilon$-quantile regret [Chaudhuri et al., 2009] is defined as the difference between the cumulative loss of the learner and that of the $\lceil \varepsilon d \rceil$-th best expert, where $\varepsilon \in [1/d, 1]$. Let $i_\varepsilon$ be the $\lceil \varepsilon d \rceil$-th best expert. Then, the $\varepsilon$-quantile regret is calculated as:

$$\mathrm{Reg}_\varepsilon = \sum_{t=1}^{T} \langle p_t, \ell_t \rangle - \sum_{t=1}^{T} \ell_{t,i_\varepsilon}.$$

Negrea et al. [2021] prove the minimax bound for $\varepsilon$-quantile regret to be $\mathcal{O}\left(\sqrt{T \log \frac{1}{\varepsilon}}\right)$. We show that our Algorithm 2 achieves a near-optimal rate for quantile regret using Theorem 4.1 and ideas similar to Remark 9.16 of [Orabona, 2019].

**Theorem B.3.** *For all $\varepsilon \in [1/d, 1]$, Algorithm 2 guarantees*

$$\text{Reg}_\varepsilon \leq \mathcal{O}\left(\sqrt{T \log \frac{1}{\varepsilon}} + \sqrt{T \log \log d}\right).$$

*Proof of Theorem B.3.* We order the $d$ experts in increasing order of their cumulative losses. Let $q_\varepsilon$ be the probability distribution over these $d$ experts with probability mass $\frac{1}{\lceil \varepsilon d \rceil}$ on the first $\lceil \varepsilon d \rceil$ experts in the ordered list and $0$ on the remaining experts. Let $\phi_\varepsilon = \mathbf{1} q_\varepsilon^\top$, i.e., each row of $\phi_\varepsilon$ is $q_\varepsilon^\top$. We can also express it as $\sum_{i=1}^d q_{\varepsilon,i}(\mathbf{1} e_i^\top)$, a convex combination of binary swap matrices. Therefore, it can be regarded as a distribution over the set $\{\mathbf{1} e_1^\top, \ldots, \mathbf{1} e_d^\top\}$.

Next, we consider the probability assigned by $\pi$ to the swap matrices in this set. For all $i \in [d]$, $\pi(\mathbf{1} e_i^\top) \geq \frac{1}{2d} \pi_{\psi^i}(\mathbf{1} e_i^\top) = \frac{1}{2d}\left(1 - \frac{1}{d}\right)^d \geq \frac{1}{8d}$ for $d \geq 2$. Now, we show that comparing against $\phi_\varepsilon$ gives us the desired bound for $\varepsilon$-quantile regret:

$$
\begin{aligned}
\text{Reg}_\varepsilon &= \sum_{t=1}^T \langle p_t, \ell_t \rangle - \sum_{t=1}^T \ell_{t,i_\varepsilon} \\
&\leq \sum_{t=1}^T \langle p_t - q_\varepsilon, \ell_t \rangle \\
&= \sum_{t=1}^T \langle \phi_t(p_t) - \phi_\varepsilon(p_t), \ell_t \rangle && \text{(since for all } p \in \Delta(d), \phi_\varepsilon(p) = q_\varepsilon) \\
&\leq \mathcal{O}\left(\sqrt{T \text{KL}(q_\varepsilon, \pi)} + \sqrt{T \log \log d}\right) && \text{(by Theorem 4.1)} \\
&= \mathcal{O}\left(\sqrt{T \sum_{i=1}^d q_{\varepsilon,i} \log \frac{q_{\varepsilon,i}}{\pi\left(\mathbf{1} e_i^\top\right)}} + \sqrt{T \log \log d}\right) \\
&&\text{(only the elements in } \{\mathbf{1} e_1^\top, \ldots, \mathbf{1} e_d^\top\} \text{ have non-zero probability mass in } q_\varepsilon) \\
&\leq \mathcal{O}\left(\sqrt{T \log \frac{8d}{\lceil \varepsilon d \rceil}} + \sqrt{T \log \log d}\right) && \text{(since } q_{\varepsilon,i} = \frac{1}{\lceil \varepsilon d \rceil} \text{ for all } i \in [d]) \\
&\leq \mathcal{O}\left(\sqrt{T \log \frac{1}{\varepsilon}} + \sqrt{T \log \log d}\right).
\end{aligned}
$$

This completes the proof. □

## B.3 Kernelized MWU

In this section, we first prove Theorem 4.3, and then discuss how to further speed up Algorithm 3.

### B.3.1 Proof of Theorem 4.3

*Proof.* The kernel function (Definition 4.2) used in Algorithm 3 can be computed as follows:

$$
\begin{aligned}
K(B, A) &= \sum_{\phi \in \Phi_b} \pi(\phi) \prod_{i,j \in [d]: \phi_{ij}=1} B_{ij} A_{ij} \\
&= \frac{1}{2d} \sum_{k=1}^d \sum_{\phi \in \Phi_b} \prod_{i,j \in [d]: \phi_{ij}=1} \psi_{ij}^k B_{ij} A_{ij} + \frac{1}{2} \sum_{\phi \in \Phi_b} \prod_{i,j \in [d]: \phi_{ij}=1} \psi_{ij}^{d+1} B_{ij} A_{ij} \\
&&\text{(from Eq. (1))} \\
&= \frac{1}{2d} \sum_{k=1}^d \prod_{i=1}^d \sum_{j=1}^d \psi_{ij}^k B_{ij} A_{ij} + \frac{1}{2} \prod_{i=1}^d \sum_{j=1}^d \psi_{ij}^{d+1} B_{ij} A_{ij}.
\end{aligned}
$$

Thus, it takes $\mathcal{O}(d^3)$ time to evaluate it.

To prove the equivalence between Algorithm 1 and Algorithm 3, we denote $J = \mathbf{1}\mathbf{1}^\top, \overline{J}_{ij} = J - e_i e_j^\top$, and $l_{t,\phi} = \langle \phi, p_t \ell_t^\top \rangle$. With some abuse in notation, for $\phi \in \Phi_b$ and $i \in [d]$, we denote by $\phi(i)$ the unique index $j \in [d]$ such that $\phi_{ij} = 1$.

According to MWU (Algorithm 1), $q_t(\phi) = \frac{\pi(\phi) \exp\left(-\eta \sum_{\tau=1}^{t-1} l_{\tau,\phi}\right)}{\sum_{\phi' \in \Phi_b} \pi(\phi') \exp\left(-\eta \sum_{\tau=1}^{t-1} l_{\tau,\phi'}\right)}$, for all $\phi \in \Phi_b$. From Kernelized MWU (Algorithm 3), we have

$$K(B_t, J) = \sum_{\phi \in \Phi_b} \pi(\phi) \prod_{i,j \in [d]:\phi_{ij}=1} (B_t)_{ij} J_{ij}$$

$$= \sum_{\phi \in \Phi_b} \pi(\phi) \prod_{i,j \in [d]:\phi_{ij}=1} \exp\left(-\eta \sum_{\tau=1}^{t-1} p_{\tau,i} \ell_{\tau,j}\right)$$

and

$$K(B_t, \overline{J}_{ij}) = \sum_{\phi \in \Phi_b} \pi(\phi) \prod_{u,v \in [d]:\phi_{uv}=1} (B_t)_{uv} \left(\overline{J}_{ij}\right)_{uv}$$

$$= \sum_{\phi \in \Phi_b:\phi(i)\neq j} \pi(\phi) \prod_{u,v \in [d]:\phi_{uv}=1} \exp\left(-\eta \sum_{\tau=1}^{t-1} p_{\tau,u} \ell_{\tau,v}\right).$$

So, we have

$$K(B_t, J) - K(B_t, \overline{J}_{ij}) = \sum_{\phi \in \Phi_b:\phi(i)=j} \pi(\phi) \prod_{u,v \in [d]:\phi_{uv}=1} \exp\left(-\eta \sum_{\tau=1}^{t-1} p_{\tau,u} \ell_{\tau,v}\right)$$

$$= \sum_{\phi \in \Phi_b:\phi(i)=j} \pi(\phi) \exp\left(-\eta \sum_{\tau=1}^{t-1} l_{\tau,\phi}\right)$$

$$= \sum_{\phi \in \Phi_b} \pi(\phi) \exp\left(-\eta \sum_{\tau=1}^{t-1} l_{\tau,\phi}\right) \phi_{ij}.$$

Therefore, for all $i, j \in [d]$, we get,

$$(\phi_t)_{ij} = \frac{K(B_t, J) - K(B_t, \overline{J}_{ij})}{K(B_t, J)}$$

$$= \frac{\sum_{\phi \in \Phi_b} \pi(\phi) \exp\left(-\eta \sum_{\tau=1}^{t-1} l_{\tau,\phi}\right) \phi_{ij}}{\sum_{\phi \in \Phi_b} \pi(\phi) \exp\left(-\eta \sum_{\tau=1}^{t-1} l_{\tau,\phi}\right)}$$

$$= \sum_{\phi \in \Phi_b} q_t(\phi) \phi_{ij},$$

giving us the required equivalence: $\phi_t = \sum_{\phi \in \Phi_b} q_t(\phi) \cdot \phi = \mathbb{E}_{\phi \sim q_t}[\phi]$. $\qquad \square$

### B.3.2   More Efficient Implementation of Algorithm 3

Based on Theorem B.4, each iteration of Algorithm 3 can be implemented in $\mathcal{O}(d^5)$ time. However, we show below that by reusing intermediate statistics and expanding the terms $\psi^k$ using Eq. (2), this can be improved to $\mathcal{O}(d^2)$.

**Theorem B.4.** *Algorithm 6 outputs the same $\phi_t$ as Algorithm 3 for all $t \in [T]$ and has a time complexity of $\mathcal{O}\left(d^2\right)$ per iteration.*

*Proof.* Similarly, we denote $J = \mathbf{1}\mathbf{1}^\top$ and $\overline{J}_{ij} = J - e_i e_j^\top$. With some abuse in notation, for $\phi \in \Phi_b$ and $i \in [d]$, we denote by $\phi(i)$ the unique index $j \in [d]$ such that $\phi_{ij} = 1$. Direct calculation shows

---
**Algorithm 6** Faster Kernelized MWU with non-uniform prior
---
**Input:** learning rate $\eta > 0$.
**Initialize:** $L_0 \in \mathbb{R}^{d \times d}$ as the all-zero matrix.
**for** $t = 1, 2, \cdots, T$ **do**

> Compute the quantities $V_t \in \mathbb{R}^{d \times d}$, $c_t \in \mathbb{R}$, $C_t \in \mathbb{R}^{d \times (d+1)}$, and $S_t \in \mathbb{R}^d$ as follows:
>
> $$(V_t)_{ik} = \frac{\exp(-\eta(L_{t-1})_{ik})}{\sum_{j=1}^d \exp(-\eta(L_{t-1})_{ij})}, \ \forall \, i, k \in [d]$$
>
> $$c_t = \frac{1}{d} \sum_{k=1}^d \prod_{i=1}^d \left( (V_t)_{ik} + \frac{1}{d(d-2)} \right) + \prod_{i=1}^d \left( (V_t)_{ii} + \frac{1}{d(d-2)} \right)$$
>
> $$(C_t)_{ik} = \begin{cases} \prod_{u \neq i} \left( (V_t)_{uk} + \frac{1}{d(d-2)} \right), & k \in [d], \\ \prod_{u \neq i} \left( (V_t)_{uu} + \frac{1}{d(d-2)} \right), & k = d+1, \end{cases} \quad \forall \, i \in [d]. \tag{9}$$
>
> $$(S_t)_i = \sum_{k=1}^d (C_t)_{ik}, \ \forall \, i \in [d]$$
>
> Compute $\phi_t$ as:
>
> $$(\phi_t)_{ij} = \frac{(V_t)_{ij}(C_t)_{ij}}{c_t d} + \frac{(V_t)_{ij}(S_t)_i}{c_t d^2(d-2)} + \left( \frac{1}{d(d-2)} + \mathbb{1}\{i = j\} \right) \frac{(V_t)_{ij}(C_t)_{i,d+1}}{c_t}, \ \forall \, i, j \in [d] \tag{10}$$
>
> Receive loss matrix $p_t \ell_t^\top$, and update $L_t = L_{t-1} + p_t \ell_t^\top$.
---

that

$$K(B_t, J) = \sum_{\phi \in \Phi_b} \pi(\phi) \prod_{i,j \in [d]: \phi_{ij} = 1} (B_t)_{ij}(J)_{ij}$$

$$= \sum_{\phi \in \Phi_b} \left( \frac{1}{2d} \sum_{k=1}^d \prod_{i,j \in [d]: \phi_{ij}=1} \psi_{ij}^k + \frac{1}{2} \prod_{i,j \in [d]: \phi_{ij}=1} \psi_{ij}^{d+1} \right) \prod_{i,j \in [d]: \phi_{ij}=1} \exp\left( -\eta(L_{t-1})_{ij} \right)$$

$$= \frac{1}{2d} \sum_{k=1}^d \prod_{i=1}^d \sum_{j=1}^d \psi_{ij}^k \exp\left( -\eta(L_{t-1})_{ij} \right) + \frac{1}{2} \prod_{i=1}^d \sum_{j=1}^d \psi_{ij}^{d+1} \exp\left( -\eta(L_{t-1})_{ij} \right),$$

where $L_{t-1}$ is defined in Algorithm 6.

Using the definition of $\psi^k$ in Eq. (2), for $i, k \in [d]$, we have

$$\sum_{j=1}^d \psi_{ij}^k \exp(-\eta(L_{t-1})_{ij}) = \left( 1 - \frac{1}{d} \right) \exp(-\eta(L_{t-1})_{ik}) + \frac{1}{d(d-1)} \sum_{j \neq k} \exp(-\eta(L_{t-1})_{ij})$$

$$= \left( \frac{d-2}{d-1} \right) \exp(-\eta(L_{t-1})_{ik}) + \frac{1}{d(d-1)} \sum_{j=1}^d \exp(-\eta(L_{t-1})_{ij})$$

$$= \left( \sum_{j=1}^d \exp(-\eta(L_{t-1})_{ij}) \right) \left( \left( \frac{d-2}{d-1} \right) \frac{\exp(-\eta(L_{t-1})_{ik})}{\sum_{j=1}^d \exp(-\eta(L_{t-1})_{ij})} + \frac{1}{d(d-1)} \right)$$

$$= \left( \frac{d-2}{d-1} \right) \left( \sum_{j=1}^d \exp(-\eta(L_{t-1})_{ij}) \right) \left( (V_t)_{ik} + \frac{1}{d(d-2)} \right).$$

$$\tag{11}$$

Similarly using the definition of $\psi^{d+1}$ in Eq. (2), we have

$$\sum_{j=1}^{d} \psi_{ij}^{d+1} \exp(-\eta(L_{t-1})_{ij}) = \left(\frac{d-2}{d-1}\right) \left(\sum_{j=1}^{d} \exp(-\eta(L_{t-1})_{ij})\right) \left((V_t)_{ii} + \frac{1}{d(d-2)}\right). \quad (12)$$

Thus,

$$\begin{aligned}
K(B_t, J) &= \frac{1}{2d} \sum_{k=1}^{d} \left(\frac{d-2}{d-1}\right)^d \prod_{i=1}^{d} \left(\sum_{j=1}^{d} \exp(-\eta(L_{t-1})_{ij})\right) \prod_{i=1}^{d} \left((V_t)_{ik} + \frac{1}{d(d-2)}\right) \\
&\quad + \frac{1}{2}\left(\frac{d-2}{d-1}\right)^d \prod_{i=1}^{d} \left(\sum_{j=1}^{d} \exp(-\eta(L_{t-1})_{ij})\right) \prod_{i=1}^{d} \left((V_t)_{ii} + \frac{1}{d(d-2)}\right) \\
&= \frac{1}{2}\left(\frac{d-2}{d-1}\right)^d c_t \prod_{i=1}^{d} \left(\sum_{j=1}^{d} \exp(-\eta(L_{t-1})_{ij})\right).
\end{aligned}$$

Similarly, we have

$$\begin{aligned}
K(B_t, \overline{J}_{ij}) &= \sum_{\phi \in \Phi_b} \pi(\phi) \prod_{u,v \in [d]:\phi_{uv}=1} (B_t)_{uv}(\overline{J}_{ij})_{uv} \\
&= \sum_{\phi \in \Phi_b} \left(\frac{1}{2d} \sum_{k=1}^{d} \prod_{u,v \in [d]:\phi_{uv}=1} \psi_{uv}^{k} + \frac{1}{2} \prod_{u,v \in [d]:\phi_{uv}=1} \psi_{uv}^{d+1}\right) \prod_{u,v \in [d]:\phi_{uv}=1} (B_t)_{uv}(\overline{J}_{ij})_{uv} \\
&= \sum_{\phi:\phi(i)\neq j} \left(\frac{1}{2d} \sum_{k=1}^{d} \prod_{u,v \in [d]:\phi_{uv}=1} \psi_{uv}^{k} + \frac{1}{2} \prod_{u,v \in [d]:\phi_{uv}=1} \psi_{uv}^{d+1}\right) \prod_{u,v \in [d]:\phi_{uv}=1} \exp\left(-\eta(L_{t-1})_{uv}\right).
\end{aligned}$$

This implies:

$$\begin{aligned}
&K(B_t, J) - K(B_t, \overline{J}_{ij}) \\
&= \sum_{\phi:\phi(i)=j} \left(\frac{1}{2d} \sum_{k=1}^{d} \prod_{u,v \in [d]:\phi_{uv}=1} \psi_{uv}^{k} + \frac{1}{2} \prod_{u,v \in [d]:\phi_{uv}=1} \psi_{uv}^{d+1}\right) \prod_{u,v \in [d]:\phi_{uv}=1} \exp\left(-\eta(L_{t-1})_{uv}\right) \\
&= \frac{1}{2d} \sum_{k=1}^{d} \psi_{ij}^{k} \exp\left(-\eta(L_{t-1})_{ij}\right) \sum_{\phi:\phi(i)=j} \prod_{u\neq i:\phi_{uv}=1} \psi_{uv}^{k} \exp\left(-\eta(L_{t-1})_{uv}\right) \\
&\quad + \frac{1}{2}\psi_{ij}^{d+1} \exp\left(-\eta(L_{t-1})_{ij}\right) \sum_{\phi:\phi(i)=j} \prod_{u\neq i:\phi_{uv}=1} \psi_{uv}^{d+1} \exp\left(-\eta(L_{t-1})_{uv}\right) \\
&= \frac{1}{2d} \sum_{k=1}^{d} \psi_{ij}^{k} \exp\left(-\eta(L_{t-1})_{ij}\right) \prod_{u\neq i} \sum_{v=1}^{d} \psi_{uv}^{k} \exp\left(-\eta(L_{t-1})_{uv}\right) \\
&\quad + \frac{1}{2}\psi_{ij}^{d+1} \exp\left(-\eta(L_{t-1})_{ij}\right) \prod_{u\neq i} \sum_{v=1}^{d} \psi_{uv}^{d+1} \exp\left(-\eta(L_{t-1})_{uv}\right) \\
&= \frac{1}{2d} \sum_{k=1}^{d} \psi_{ij}^{k} \exp\left(-\eta(L_{t-1})_{ij}\right) \prod_{u\neq i} \left(\frac{d-2}{d-1}\right) \left(\sum_{v=1}^{d} \exp(-\eta(L_{t-1})_{uv})\right) \left((V_t)_{uk} + \frac{1}{d(d-2)}\right) \\
&\quad + \frac{1}{2}\psi_{ij}^{d+1} \exp\left(-\eta(L_{t-1})_{ij}\right) \prod_{u\neq i} \left(\frac{d-2}{d-1}\right) \left(\sum_{v=1}^{d} \exp(-\eta(L_{t-1})_{uv})\right) \left((V_t)_{uu} + \frac{1}{d(d-2)}\right)
\end{aligned}$$

$$\text{(from Eq. (11) and Eq. (12))}$$

$$= \frac{1}{2d} \sum_{k=1}^{d} \psi_{ij}^{k} \frac{\exp\left(-\eta(L_{t-1})_{ij}\right)}{\sum_{v=1}^{d} \exp\left(-\eta(L_{t-1})_{iv}\right)} \left(\frac{d-2}{d-1}\right)^{d-1} \left(\prod_{u=1}^{d} \sum_{v=1}^{d} \exp(-\eta(L_{t-1})_{uv})\right)(C_t)_{ik}$$

$$+ \frac{1}{2} \psi_{ij}^{d+1} \frac{\exp\left(-\eta(L_{t-1})_{ij}\right)}{\sum_{v=1}^{d} \exp\left(-\eta(L_{t-1})_{iv}\right)} \left(\frac{d-2}{d-1}\right)^{d-1} \left(\prod_{u=1}^{d} \sum_{v=1}^{d} \exp(-\eta(L_{t-1})_{uv})\right)(C_t)_{i,d+1}$$

(from Eq. (9))

$$= \frac{1}{2} \left(\frac{d-2}{d-1}\right)^{d-1} \left(\prod_{u=1}^{d} \sum_{v=1}^{d} \exp(-\eta(L_{t-1})_{uv})\right)(V_t)_{ij} \left(\frac{1}{d} \sum_{k=1}^{d} \psi_{ij}^{k}(C_t)_{ik} + \psi_{ij}^{d+1}(C_t)_{i,d+1}\right)$$

(from Eq. (9))

Therefore, we get

$$\frac{K(B_t, J) - K(B_t, \overline{J}_{ij})}{K(B_t, J)} = \frac{d-1}{c_t(d-2)}(V_t)_{ij} \left(\frac{1}{d} \sum_{k=1}^{d} \psi_{ij}^{k}(C_t)_{ik} + \psi_{ij}^{d+1}(C_t)_{i,d+1}\right)$$

where the left-hand side is how $(\phi_t)_{ij}$ is defined in Algorithm 3.

Finally, we consider the following two cases.
Case 1: for $j \neq i$, we have

$$(\phi_t)_{ij} = \frac{d-1}{c_t d(d-2)}(V_t)_{ij} \left(\left(1 - \frac{1}{d}\right)(C_t)_{ij} + \frac{1}{d(d-1)} \sum_{k \neq j}(C_t)_{ik}\right)$$

$$+ \frac{d-1}{c_t(d-2)}(V_t)_{ij} \frac{1}{d(d-1)}(C_t)_{i,d+1}$$

(from Eq. (2))

$$= \frac{(V_t)_{ij}(C_t)_{ij}}{c_t d} + \frac{(V_t)_{ij}(S_t)_i}{c_t d^2(d-2)} + \frac{1}{d(d-2)} \frac{(V_t)_{ij}(C_t)_{i,d+1}}{c_t}.$$

(from Eq. (9))

Case 2: for $j = i$, we have

$$(\phi_t)_{ij} = \frac{d-1}{c_t d(d-2)}(V_t)_{ij} \left(\left(1 - \frac{1}{d}\right)(C_t)_{ij} + \frac{1}{d(d-1)} \sum_{k \neq j}(C_t)_{ik}\right)$$

$$+ \frac{d-1}{c_t(d-2)}(V_t)_{ij} \left(\frac{d-1}{d}\right)(C_t)_{i,d+1}$$

(from Eq. (2))

$$= \frac{(V_t)_{ij}(C_t)_{ij}}{c_t d} + \frac{(V_t)_{ij}(S_t)_i}{c_t d^2(d-2)} + \frac{(d-1)^2}{d(d-2)} \cdot \frac{(V_t)_{ij}(C_t)_{i,d+1}}{c_t}.$$

(from Eq. (9))

Combining the cases above gives us how $(\phi_t)_{ij}$ is defined in Algorithm 6, establishing the claimed equivalence.

To calculate the time complexity of computing $\phi_t$, note that $V_t$ can be calculated in $\mathcal{O}\left(d^2\right)$ time. Given $V_t$, computing $c_t$ takes another $\mathcal{O}\left(d^2\right)$ time. To compute $C_t$, we can first compute $\prod_{u=1}^{d}\left((V_t)_{uk} + \frac{1}{d(d-2)}\right), \ \forall \ k \in [d]$ and $\prod_{u=1}^{d}\left((V_t)_{uu} + \frac{1}{d(d-2)}\right)$ in $\mathcal{O}\left(d^2\right)$ time. Then, these values can be used to calculate the matrix $C_t$ in $\mathcal{O}\left(d^2\right)$ time because we can compute each entry of $C_t$ in constant time. With $C_t$, $S_t$ can also be computed in $\mathcal{O}\left(d^2\right)$ time. Therefore, computing $\phi_t$ takes $\mathcal{O}\left(d^2\right)$ time. $\qquad \square$

## C   Omitted Details in Section 5

First, we include the meta MWU algorithm discussed in Section 5 in Algorithm 7, which uses a base algorithm shown in Algorithm 8. We use $\mathcal{U} \triangleq [d + 1] \times [M]$ for notational convenience, where $M = 2\lceil \log_2 d \rceil$. We now show the guarantee for our proposed prior-aware BM-reduction (Algorithm 8).

**Theorem C.1.** *Suppose that $\pi_\psi$ is a $\psi$-induced distribution as defined in Definition 3.1. Then Algorithm 8 with prior $\pi_\psi$, learning rate $\eta > 0$, and* SubAlg *being MWU (Algorithm 5 with $\mathcal{A} = [d]$) guarantees $\sum_{t=1}^{T} \left\langle \phi_t - \phi, p_t \ell_t^\top \right\rangle \leq \frac{\log \frac{1}{\pi_\psi(\phi)}}{\eta} + \eta T$ for all $\phi \in \Phi_b$.*

---

**Algorithm 7** Meta MWU Algorithm for Learning Multiple BM-Reductions

---

1 **Initialization:** Set learning rate $\eta = \sqrt{\frac{\log((d+1)\cdot 2\lceil \log_2 d\rceil)}{T}}$ and $w_1 = \frac{1}{|\mathcal{U}|}\mathbf{1} \in \Delta(\mathcal{U})$, where $\mathcal{U} = [d+1] \times [M]$; initialize $|\mathcal{U}|$ base-learner $\mathcal{B}_{k,h}$, $(k,h) \in \mathcal{U}$, where $\mathcal{B}_{k,h}$ is an instance of Algorithm 8 with prior $\psi^k$, learning rate $\eta_h = \sqrt{2^h/T}$, and subroutine SubAlg being MWU (Algorithm 5 with $\mathcal{A} = [d]$).

   **for** $t = 1, 2, \cdots, T$ **do**

2   Receive $\phi_t^{k,h} \in \mathcal{S}$ from $\mathcal{B}_{k,h}$ for each $(k,h) \in \mathcal{U}$ and compute $\phi_t = \sum_{(k,h)\in\mathcal{U}} w_{t,k,h}\phi_t^{k,h}$.

3   Play the stationary distribution $p_t$ of $\phi_t$ (that is, $p_t = \phi_t(p_t)$) and receive loss $\ell_t$.

4   Update $w_{t+1}$ such that $w_{t+1,k,h} \propto w_{t,k,h}\exp(-\eta\ell_{t,k,h}^w)$ where $\ell_{t,k,h}^w = \langle \phi_t^{k,h}, p_t\ell_t^\top\rangle$.

5   Send loss matrix $p_t\ell_t^\top$ to $\mathcal{B}_{k,h}$ for each $(k,h) \in \mathcal{U}$.

---

---

**Algorithm 8** Prior-Aware BM-Reduction

---

**Input:** a prior $\psi \in \mathcal{S}$, a learning rate $\eta > 0$, and an external regret minimization subroutine SubAlg.

1 **Initialize:** $d$ instances of SubAlg, denoted by $\mathsf{SubAlg}_1, \ldots, \mathsf{SubAlg}_d$, where $\mathsf{SubAlg}_k$ uses learning rate $\eta$ and prior distribution $\psi_{k:} \in \Delta(d)$.

   **for** $t = 1, 2 \ldots, T$ **do**

2   Propose $\phi_t \in \mathcal{S}$ where the $k$-th row $\phi_{t,k:} \in \Delta(d)$ is the output of $\mathsf{SubAlg}_k$.

3   Receive a loss matrix $p_t\ell_t^\top$ and send the $k$-th row to $\mathsf{SubAlg}_k$ for each $k \in [d]$.

---

*Proof of Theorem C.1.* First we decompose $\sum_{t=1}^T \langle \phi_t - \phi, p_t\ell_t^\top\rangle$ as $\sum_{t=1}^T \sum_{i=1}^d \langle \phi_{t,i:} - \phi_{i:}, p_{t,i}\ell_t\rangle$. For each $i$, based on the algorithm and Lemma B.1, we have

$$\sum_{i=1}^d \langle \phi_{t,i:} - \phi_{i:}, p_{t,i}\ell_t\rangle \leq \frac{\log\frac{1}{\psi_{i,\phi(i)}}}{\eta} + \eta\sum_{t=1}^T p_{t,i}$$

where $\phi(i)$ denotes the unique index $j$ such that $\phi_{ij} = 1$. Noting that $\sum_{i=1}^d \psi_{i,\phi(i)}$ is exactly $\pi_\psi(\phi)$ by definition, we have thus proven

$$\sum_{t=1}^T \langle \phi_t - \phi, p_t\ell_t^\top\rangle \leq \sum_{i=1}^d \left(\frac{\log\frac{1}{\psi_{i,\phi(i)}}}{\eta} + \eta\sum_{t=1}^T p_{t,i}\right) = \frac{\log\frac{1}{\pi_\psi(\phi)}}{\eta} + \eta T.$$

$\square$

Next, we provide the proof for the adaptive $\Phi$-regret achieved by Algorithm 7.

*Proof of Theorem 5.1.* Since $p_t$ is a stationary distribution of $\phi_t$, we have $\mathrm{Reg}(\phi) = \sum_{t=1}^T \langle p_t - \phi(p_t), \ell_t\rangle = \sum_{t=1}^T \langle \phi_t(p_t) - \phi(p_t), \ell_t\rangle = \sum_{t=1}^T \langle \phi_t - \phi, p_t\ell_t^\top\rangle$. Using the definition of $\phi_t$ and $\ell_t^w$ from Algorithm 7, for any $(k,h) \in \mathcal{U}$, we decompose $\mathrm{Reg}(\phi)$ as

$$\begin{aligned}
\mathrm{Reg}(\phi) &= \sum_{t=1}^T \langle \phi_t - \phi, p_t\ell_t^\top\rangle \\
&= \sum_{t=1}^T \left\langle \sum_{(k,h)\in\mathcal{U}} w_{t,k,h}\phi_t^{k,h} - \phi, p_t\ell_t^\top\right\rangle \\
&= \sum_{t=1}^T \langle w_t - e_{k,h}, \ell_t^w\rangle + \sum_{t=1}^T \left\langle \phi_t^{k,h} - \phi, p_t\ell_t^\top\right\rangle.
\end{aligned}$$

Applying Lemma B.1, the first term can be bounded as

$$\sum_{t=1}^T \langle w_t - e_{k,h}, \ell_t^w\rangle \leq 2\sqrt{T\log\left((d+1)\cdot 2\lceil\log_2 d\rceil\right)} \leq 4\sqrt{T\log d}. \tag{13}$$

For the second term, by Theorem C.1, it holds that

$$\sum_{t=1}^{T}\langle \phi_t^{k,h} - \phi, p_t\ell_t^{\top}\rangle \leq \frac{\log \frac{1}{\pi_{\psi^k}(\phi)}}{\eta_h} + \eta_h T. \tag{14}$$

Summing up Eq. (13) and Eq. (14), we can bound $\mathrm{Reg}(\phi)$ as

$$\mathrm{Reg}(\phi) \leq \frac{\log \frac{1}{\pi_{\psi^k}(\phi)}}{\eta_h} + \eta_h T + 4\sqrt{T\log d}.$$

Since the above inequality holds for all $k \in [d+1]$, we have

$$\begin{aligned}\mathrm{Reg}(\phi) &\leq \min_{k\in[d+1]} \frac{\log \frac{1}{\pi_{\psi^k}(\phi)}}{\eta_h} + \eta_h T + 4\sqrt{T\log d} \\ &= \frac{\log \frac{1}{\max_{k\in[d+1]}\pi_{\psi^k}(\phi)}}{\eta_h} + \eta_h T + 4\sqrt{T\log d} \\ &\leq \frac{\log \frac{1}{\pi(\phi)}}{\eta_h} + \eta_h T + 4\sqrt{T\log d}, \end{aligned} \tag{15}$$

by the definition of $\pi$ (Definition 3.2). It is clear that Eq. (15) attains its minimum when $\eta_h$ is $\sqrt{\frac{\log \frac{1}{\pi(\phi)}}{T}}$. Similar to Lemma B.2, we now show that there exists $h^\star$ such that $\eta_{h^\star}$ is close to this optimum. Since $\psi_{ij}^k \geq \frac{1}{d^2}$ for all $k \in [d+1]$ and $i, j \in [d]$, it holds that

$$\min_h 2^h = 2 \leq \max\left\{\log\frac{1}{\pi(\phi)}, 2\right\} \leq d^2 = 2^{2\log_2 d} \leq \max_h 2^h$$

Therefore, there exists $h^\star$ such that

$$2^{h^\star} \leq \max\left\{\log\frac{1}{\pi(\phi)}, 2\right\} \leq 2^{h^\star+1},$$

and thus

$$\frac{\log\frac{1}{\pi(\phi)}}{\eta_{h^\star}} + \eta_{h^\star}T = \log\frac{1}{\pi(\phi)}\cdot\sqrt{\frac{T}{2^{h^\star}}} + \sqrt{\frac{2^{h^\star}}{T}}T \leq 3\sqrt{T\log\frac{1}{\pi(\phi)}} + 2\sqrt{T}.$$

Substituting it into Eq. (15) (by picking $h = h^\star$), we have $\mathrm{Reg}(\phi) \leq 3\sqrt{T\log\frac{1}{\pi(\phi)}} + 2\sqrt{T} + 4\sqrt{T\log d} = \mathcal{O}\left(\sqrt{T\log\frac{1}{\pi(\phi)}} + \sqrt{T\log d}\right)$. Therefore, Eq. (3) is satisfied with $B = \sqrt{T\log d}$. The second statement of the theorem then follows directly from Theorem 3.3. $\qquad\square$

## D   Omitted Details in Section 6

In this section, we provide the omitted details and proofs for our results in Section 6. The section is organized as follows. In Appendix D.1, we include the pseudocode for OMWU. In Appendix D.2, we introduce several important lemmas that will be useful in our analysis. Then, in Appendix D.3, we provide the full proof for Theorem 6.4. Specifically, we start with a proof sketch, showing how we utilize the nonnegative-social-external-regret property to show that the path-length of the entire learning dynamic is bounded by $\mathcal{O}(N\log d)$, followed by a full proof of Theorem 6.4. Importantly, following the notation convention introduced in Section 6, **we use superscript** $(n)$ **to denote variables associated with agent/player** $n$.

### D.1   Pseudocode for OMWU

Here, we include the pseudocode for OMWU (Algorithm 9) that is used in Algorithm 4. There are two possible outputs for Algorithm 9 at each round $t$. For base learner $\mathcal{B}_{d+2}$ in Algorithm 4, the output in round $t$ is $\phi_t \in \mathbb{R}^{d\times d}$, while for subroutines used by $\mathcal{B}_k$ for $k \in [d+1]$, the output is $p_t \in \Delta(d)$.

**Algorithm 9** OMWU

---

**Input:** learning rate $\eta > 0$; a prior distribution $\widehat{p}_1 \in \Delta(d)$.
1 **Initialize:** $\ell_0 = \mathbf{0} \in \mathbb{R}^d$.
  **for** $t = 1, 2 \dots, T$ **do**
2     Compute $p_t$ such that $p_{t,i} \propto \widehat{p}_{t,i} \exp(-\eta \ell_{t-1,i})$ for $i \in [d]$ and $\phi_t = \mathbf{1}p_t^\top \in \mathbb{R}^{d \times d}$.
3     Receive $\ell_t$ and compute $\widehat{p}_{t+1}$ such that $\widehat{p}_{t+1,i} \propto \widehat{p}_{t,i} \exp(-\eta \ell_{t,i})$ for $i \in [d]$.

---

### D.2 Auxiliary Lemmas

To analyze the performance of OMWU, we use following lemma from Syrgkanis et al. [2015].

**Lemma D.1** (Theorem 18 in [Syrgkanis et al., 2015]). *OMWU (Algorithm 9) with learning rate $\eta > 0$ guarantees that*

$$\sum_{t=1}^T \langle p_t - u, \ell_t \rangle \leq \frac{\mathrm{KL}(u, p_1)}{\eta} + \eta \sum_{t=2}^T \|\ell_t - \ell_{t-1}\|_\infty^2 - \frac{1}{8\eta} \sum_{t=2}^T \|p_t - p_{t-1}\|_1^2.$$

The next lemma shows that the loss vector difference between consecutive rounds for each agent $n$ is bounded by the sum of the strategy differences over all other agents.

**Lemma D.2.** *For any $t \in [T]$, $n \in [N]$, we have*

$$\|\ell_t^{(n)} - \ell_{t-1}^{(n)}\|_\infty^2 \leq (N-1) \sum_{j \neq n} \|p_t^{(j)} - p_{t-1}^{(j)}\|_1^2.$$

*Proof.* For any action $a \in [d]$:

$$
\begin{aligned}
|\ell_{t,a}^{(n)} - \ell_{t-1,a}^{(n)}| &= \left| \sum_{\mathbf{a}^{(-n)}} \left( \prod_{j \neq n} p_{t,a_j}^{(j)} - \prod_{j \neq n} p_{t-1,a_j}^{(j)} \right) \cdot \ell^{(n)}(a, \mathbf{a}^{(-n)}) \right| \\
&\leq \sum_{\mathbf{a}^{(-n)}} \left| \prod_{j \neq n} p_{t,a_j}^{(j)} - \prod_{j \neq n} p_{t-1,a_j}^{(j)} \right| \qquad \text{(since } |\ell^{(n)}(\mathbf{a})| \leq 1) \\
&\leq \sum_{j \neq n} \sum_{i=1}^d |p_{t,i}^{(j)} - p_{t-1,i}^{(j)}| \\
&= \sum_{j \neq n} \|p_t^{(j)} - p_{t-1}^{(j)}\|_1.
\end{aligned}
$$

Taking square on both sides, we know that

$$\|\ell_t^{(n)} - \ell_{t-1}^{(n)}\|_\infty^2 \leq \left( \sum_{j \neq n} \|p_t^{(j)} - p_{t-1}^{(j)}\|_1 \right)^2 \leq (N-1) \sum_{j \neq n} \|p_t^{(j)} - p_{t-1}^{(j)}\|_1^2.$$

$\square$

The next lemma shows how the difference between strategies in consecutive rounds is related to the stability of both the base learners and the meta learner.

**Lemma D.3.** *Suppose that every agent $n \in [N]$ applies Algorithm 4, then for all $t \geq 2$, $n \in [N]$, we have*

$$\left\| p_t^{(n)} - p_{t-1}^{(n)} \right\|_1^2 \leq 2 \sum_{k=1}^{d+2} w_{t,k}^{(n)} \left\| \widetilde{p}_t^{(n),k} - \widetilde{p}_{t-1}^{(n),k} \right\|_1^2 + 2 \left\| w_t^{(n)} - w_{t-1}^{(n)} \right\|_1^2,$$

*where $\widetilde{p}_t^{(n),k} = \phi_t^{(n),k}(p_t^{(n)})$ for each $k \in [d+2]$.*

*Proof.* Direct calculation shows that

$$\left\| p_t^{(n)} - p_{t-1}^{(n)} \right\|_1^2$$

$$= \left\| \left(\phi_t^{(n)}\right)^\top p_t^{(n)} - \left(\phi_{t-1}^{(n)}\right)^\top p_{t-1}^{(n)} \right\|_1^2 \qquad (p_t^{(n)} \text{ is stationary distribution of } \phi_t^{(n)})$$

$$= \left\| \left(\sum_{k=1}^{d+2} w_{t,k}^{(n)} \phi_t^{(n),k}\right)^\top p_t^{(n)} - \left(\sum_{k=1}^{d+2} w_{t-1,k}^{(n)} \phi_{t-1}^{(n),k}\right)^\top p_{t-1}^{(n)} \right\|_1^2 \qquad (\text{definition of } \phi_t^{(n)})$$

$$= \left\| \left(\sum_{k=1}^{d+2} w_{t,k}^{(n)} \widetilde{p}_t^{(n),k}\right) - \left(\sum_{k=1}^{d+2} w_{t-1,k}^{(n)} \widetilde{p}_{t-1}^{(n),k}\right) \right\|_1^2 \qquad (\text{definition of } \widetilde{p}_t^{(n),k})$$

$$\leq 2 \left\| \left(\sum_{k=1}^{d+2} w_{t,k}^{(n)} \widetilde{p}_t^{(n),k}\right) - \left(\sum_{k=1}^{d+2} w_{t,k}^{(n)} \widetilde{p}_{t-1}^{(n),k}\right) \right\|_1^2 + 2 \left\| \left(\sum_{k=1}^{d+2} w_{t,k}^{(n)} \widetilde{p}_{t-1}^{(n),k}\right) - \left(\sum_{k=1}^{d+2} w_{t-1,k}^{(n)} \widetilde{p}_{t-1}^{(n),k}\right) \right\|_1^2$$

$$\leq 2 \sum_{k=1}^{d+2} w_{t,k}^{(n)} \left\| \widetilde{p}_t^{(n),k} - \widetilde{p}_{t-1}^{(n),k} \right\|_1^2 + 2 \left\| w_t^{(n)} - w_{t-1}^{(n)} \right\|_1^2. \qquad (\text{Jensen's inequality})$$

$\square$

The next lemma further bounds the scale of $\|\widetilde{p}_t^{(n),k} - \widetilde{p}_{t-1}^{(n),k}\|_1^2$ with respect to the stationary distribution difference $\|p_t^{(n)} - p_{t-1}^{(n)}\|_1^2$ and the base learner's decision differences.

**Lemma D.4.** *For all* $k \in [d+2]$ *and* $n \in [N]$, $\|\widetilde{p}_t^{(n),k} - \widetilde{p}_{t-1}^{(n),k}\|_1^2 \leq 2\|p_t^{(n)} - p_{t-1}^{(n)}\|_1^2 + 2\sum_{j=1}^d \left\| \phi_{t,j:}^{(n),k} - \phi_{t-1,j:}^{(n),k} \right\|_1^2.$

*Proof.* By definition of $\widetilde{p}_t^{(n),k}$, we can bound $\|\widetilde{p}_t^{(n),k} - \widetilde{p}_{t-1}^{(n),k}\|_1^2$ as follows:

$$\|\widetilde{p}_t^{(n),k} - \widetilde{p}_{t-1}^{(n),k}\|_1^2$$

$$= \left\| (\phi_t^{(n),k})^\top p_t^{(n)} - (\phi_{t-1}^{(n),k})^\top p_{t-1}^{(n)} \right\|_1^2$$

$$\leq 2 \left\| (\phi_t^{(n),k})^\top (p_t^{(n)} - p_{t-1}^{(n)}) \right\|_1^2 + 2 \left\| (\phi_t^{(n),k} - \phi_{t-1}^{(n),k})^\top p_{t-1}^{(n)} \right\|_1^2$$

$$= 2 \left( \sum_{j=1}^d \left| \left\langle \phi_{t,:j}^{(n),k}, p_t^{(n)} - p_{t-1}^{(n)} \right\rangle \right| \right)^2 + 2 \left( \sum_{j=1}^d \left| \left\langle \phi_{t,:j}^{(n),k} - \phi_{t-1,:j}^{(n),k}, p_{t-1}^{(n)} \right\rangle \right| \right)^2$$

$$= 2 \left( \sum_{j=1}^d \left| \sum_{i=1}^d \phi_{t,ij}^{(n),k} (p_{t,i}^{(n)} - p_{t-1,i}^{(n)}) \right| \right)^2 + 2 \left( \sum_{j=1}^d \left| \sum_{i=1}^d p_{t-1,i}^{(n)} (\phi_{t,ij}^{(n),k} - \phi_{t-1,ij}^{(n),k}) \right| \right)^2$$

$$\leq 2 \left( \sum_{j=1}^d \sum_{i=1}^d \phi_{t,ij}^{(n),k} \left| p_{t,i}^{(n)} - p_{t-1,i}^{(n)} \right| \right)^2 + 2 \left( \sum_{j=1}^d \sum_{i=1}^d p_{t-1,i}^{(n)} \left| \phi_{t,ij}^{(n),k} - \phi_{t-1,ij}^{(n),k} \right| \right)^2$$

$$= 2 \left( \sum_{i=1}^d \left| p_{t,i}^{(n)} - p_{t-1,i}^{(n)} \right| \sum_{j=1}^d \phi_{t,ij}^{(n),k} \right)^2 + 2 \left( \sum_{i=1}^d p_{t-1,i}^{(n)} \left\| \phi_{t,i:}^{(n),k} - \phi_{t-1,i:}^{(n),k} \right\|_1 \right)^2$$

$$= 2 \left\| p_t^{(n)} - p_{t-1}^{(n)} \right\|_1^2 + 2 \left( \sum_{i=1}^d p_{t-1,i}^{(n)} \left\| \phi_{t,i:}^{(n),k} - \phi_{t-1,i:}^{(n),k} \right\|_1 \right)^2 \qquad (\text{since } \phi_t^{(n),k} \in \mathcal{S})$$

$$\leq 2\|p_t^{(n)} - p_{t-1}^{(n)}\|_1^2 + 2 \sum_{i=1}^d p_{t-1,i}^{(n)} \left\| \phi_{t,i:}^{(n),k} - \phi_{t-1,i:}^{(n),k} \right\|_1^2 \qquad (\text{Cauchy-Schwarz inequality})$$

$$\leq 2\|p_t^{(n)} - p_{t-1}^{(n)}\|_1^2 + 2\sum_{i=1}^{d} \left\|\phi_{t,i:}^{(n),k} - \phi_{t-1,i:}^{(n),k}\right\|_1^2,$$

which finishes the proof. □

The next lemma shows the multiplicative stability of the meta learner's strategy.

**Lemma D.5** (Multiplicative stability lemma). *Suppose that each player runs Algorithm 4 with* $\eta_m \leq \frac{1}{8(1+4\lambda)}$, *we have for all* $t \in [T]$, $n \in [N]$, *and* $k \in [d+2]$, $w_{t,k}^{(n)} \in [\frac{1}{2}w_{t-1,k}^{(n)}, 2w_{t-1,k}^{(n)}]$.

*Proof.* We omit the superscript $(n)$ for conciseness. By definition of $\ell_t^w$, $m_t^w$, and $c_t$, we know that $\max\{\|\ell_t^w + c_t\|_\infty, \|m_t^w + c_t\|_\infty\} \leq 1 + 4\lambda$ for all $t \in [T]$. Therefore, according to the update rule of $w_t$ and $\widehat{w}_t$, we know that

$$\exp(-1/8)w_{t,k} \leq \widehat{w}_{t,k} = \frac{w_{t,k}\exp(\eta_m(m_{t,k}^w + c_{t,k}))}{\sum_{i=1}^{d+2} w_{t,i}\exp(\eta_m(m_{t,i}^w + c_{t,i}))} \leq \exp(1/8)\cdot w_{t,k},$$

$$\exp(-1/8)\widehat{w}_{t-1,k} \leq \widehat{w}_{t,k} = \frac{\widehat{w}_{t-1,k}\exp(-\eta_m(\ell_{t-1,k}^w + c_{t-1,k}))}{\sum_{i=1}^{d+2} \widehat{w}_{t-1,i}\exp(-\eta_m(m_{t-1,i}^w + c_{t-1,i}))} \leq \exp(1/8)\cdot \widehat{w}_{t-1,k}.$$

Therefore, we know that $w_{t,k} \leq \exp(3/8)w_{t-1,k} \leq 2w_{t-1,k}$ and $w_{t,k} \geq \exp(-3/8)w_{t-1,k} \geq \frac{1}{2}w_{t-1,k}$. □

The next lemma bounds the external regret for the meta learner with respect to an arbitrary distribution over the $d+2$ base learners.

**Lemma D.6** (Meta Regret Bound). *Suppose that all players apply Algorithm 4 with* $\lambda \leq \frac{1}{4\eta_m}$. *Then, we have*

$$\sum_{t=1}^{T} \left\langle w_t^{(n)} - u, \ell_t^{(n),w}\right\rangle \leq \mathcal{O}(\lambda) + \frac{\mathrm{KL}(u, w_1^{(n)})}{\eta_m} + 2\eta_m N \sum_{t=2}^{T}\sum_{n=1}^{N} \|p_t^{(n)} - p_{t-1}^{(n)}\|_1^2$$

$$+ \lambda \sum_{t=2}^{T-1}\sum_{k=1}^{d+2} u_k\|\widetilde{p}_t^{(n),k} - \widetilde{p}_{t-1}^{(n),k}\|_1^2 - \frac{\lambda}{4}\sum_{t=2}^{T}\|p_t^{(n)} - p_{t-1}^{(n)}\|_1^2,$$

*for all agent* $n \in [N]$ *and* $u \in \Delta(d+2)$.

*Proof.* According to Lemma D.1, we know that for each $u \in \Delta(d+2)$ and $n \in [N]$,

$$\sum_{t=1}^{T} \left\langle w_t^{(n)} - u, \ell_t^{(n),w} + c_t^{(n)}\right\rangle$$

$$\leq \frac{\mathrm{KL}(u, w_1^{(n)})}{\eta_m} + \eta_m \sum_{t=2}^{T} \|\ell_t^{(n),w} - m_t^{(n),w}\|_\infty^2 - \frac{1}{8\eta_m}\sum_{t=2}^{T} \|w_t^{(n)} - w_{t-1}^{(n)}\|_1^2 \qquad \text{(Lemma D.1)}$$

$$= \frac{\mathrm{KL}(u, w_1^{(n)})}{\eta_m} + \eta_m \sum_{t=2}^{T} \max_{i\in[d+2]} \left|p_t^{(n)\top}\phi_t^{(n),i}\ell_t^{(n)} - p_{t-1}^{(n)\top}\phi_t^{(n),i}\ell_{t-1}^{(n)}\right|^2 - \frac{1}{8\eta_m}\sum_{t=2}^{T} \|w_t^{(n)} - w_{t-1}^{(n)}\|_1^2$$

$$\leq \frac{\mathrm{KL}(u, w_1^{(n)})}{\eta_m} - \frac{1}{8\eta_m}\sum_{t=2}^{T} \|w_t^{(n)} - w_{t-1}^{(n)}\|_1^2$$

$$+ 2\eta_m \sum_{t=2}^{T} \max_{k\in[d+2]} \left(\left|\left\langle p_t^{(n)} - p_{t-1}^{(n)}, \phi_t^{(n),k}\ell_t^{(n)}\right\rangle\right|^2 + 2\left|p_{t-1}^{(n)\top}\phi_t^{(n),k}\ell_t^{(n)} - p_{t-1}^{(n)\top}\phi_t^{(n),k}\ell_{t-1}^{(n)}\right|^2\right)$$

$$\leq \frac{\mathrm{KL}(u, w_1^{(n)})}{\eta_m} + \eta_m \sum_{t=2}^{T} \max_{i\in[d+2]} \left(2\left\|p_t^{(n)\top}\phi_t^{(n),i} - p_{t-1}^{(n)\top}\phi_t^{(n),i}\right\|_1^2 + 2\left\|\ell_t^{(n)} - \ell_{t-1}^{(n)}\right\|_\infty^2\right)$$

$$- \frac{1}{8\eta_m}\sum_{t=2}^{T} \|w_t^{(n)} - w_{t-1}^{(n)}\|_1^2 \qquad \text{(using Hölder's inequality)}$$

$$\leq \frac{\mathrm{KL}(u, w_1^{(n)})}{\eta_m} + 2\eta_m(N-1)\sum_{t=2}^{T}\sum_{n=1}^{N}\|p_t^{(n)} - p_{t-1}^{(n)}\|_1^2 - \frac{1}{8\eta_m}\sum_{t=2}^{T}\|w_t^{(n)} - w_{t-1}^{(n)}\|_1^2,$$

where the last inequality uses Lemma D.2. Recall the definition $c_{t,k}^{(n)} = \lambda\|(\phi_{t-1}^{(n),k})^\top p_{t-1}^{(n)} - (\phi_{t-2}^{(n),k})^\top p_{t-2}^{(n)}\|_1^2 = \lambda\|\widetilde{p}_{t-1}^{(n),k} - \widetilde{p}_{t-2}^{(n),k}\|_1^2$ for $t \geq 3$ and $c_{t,k}^{(n)} = 0$ for $t \in \{1,2\}$, we can further upper bound the meta regret as follows:

$$\sum_{t=1}^{T}\left\langle w_t^{(n)} - u, \ell_t^{(n),w}\right\rangle$$

$$\leq \frac{\mathrm{KL}(u, w_1^{(n)})}{\eta_m} + 2\eta_m(N-1)\sum_{t=2}^{T}\sum_{n=1}^{N}\|p_t^{(n)} - p_{t-1}^{(n)}\|_1^2 - \frac{1}{8\eta_m}\sum_{t=2}^{T}\|w_t^{(n)} - w_{t-1}^{(n)}\|_1^2$$
$$- \lambda\sum_{t=3}^{T}\sum_{k=1}^{d+2}w_{t,k}^{(n)}\|\widetilde{p}_{t-1}^{(n),k} - \widetilde{p}_{t-2}^{(n),k}\|_1^2 + \lambda\sum_{t=3}^{T}\sum_{k=1}^{d+2}u_k\|\widetilde{p}_{t-1}^{(n),k} - \widetilde{p}_{t-2}^{(n),k}\|_1^2$$

$$\leq \frac{\mathrm{KL}(u, w_1^{(n)})}{\eta_m} + 2\eta_m N\sum_{t=2}^{T}\sum_{n=1}^{N}\|p_t^{(n)} - p_{t-1}^{(n)}\|_1^2 + \lambda\sum_{t=3}^{T}\sum_{k=1}^{d+2}u_k\|\widetilde{p}_{t-1}^{(n),k} - \widetilde{p}_{t-2}^{(n),k}\|_1^2$$
$$- \frac{1}{8\eta_m}\sum_{t=2}^{T}\|w_t^{(n)} - w_{t-1}^{(n)}\|_1^2 - \frac{\lambda}{2}\sum_{t=3}^{T}\sum_{k=1}^{d+2}w_{t-1,k}^{(n)}\|\widetilde{p}_{t-1}^{(n),k} - \widetilde{p}_{t-2}^{(n),k}\|_1^2$$
$$\hspace{8cm}(w_{t-1,k}^{(n)} \leq 2w_{t,k}^{(n)} \text{ using Lemma D.5})$$

$$= \mathcal{O}(\lambda) + \frac{\mathrm{KL}(u, w_1^{(n)})}{\eta_m} + 2\eta_m N\sum_{t=2}^{T}\sum_{n=1}^{N}\|p_t^{(n)} - p_{t-1}^{(n)}\|_1^2 + \lambda\sum_{t=3}^{T}\sum_{k=1}^{d+2}u_k\|\widetilde{p}_{t-1}^{(n),k} - \widetilde{p}_{t-2}^{(n),k}\|_1^2$$
$$- \frac{1}{8\eta_m}\sum_{t=2}^{T}\|w_t^{(n)} - w_{t-1}^{(n)}\|_1^2 - \frac{\lambda}{2}\sum_{t=2}^{T}\sum_{k=1}^{d+2}w_{t,k}^{(n)}\|\widetilde{p}_t^{(n),k} - \widetilde{p}_{t-1}^{(n),k}\|_1^2$$

$$\leq \mathcal{O}(\lambda) + \frac{\mathrm{KL}(u, w_1^{(n)})}{\eta_m} + 2\eta_m N\sum_{t=2}^{T}\sum_{n=1}^{N}\|p_t^{(n)} - p_{t-1}^{(n)}\|_1^2 + \lambda\sum_{t=3}^{T}\sum_{k=1}^{d+2}u_k\|\widetilde{p}_{t-1}^{(n),k} - \widetilde{p}_{t-2}^{(n),k}\|_1^2$$
$$- \min\left\{\frac{1}{16\eta_m}, \frac{\lambda}{4}\right\}\sum_{t=2}^{T}\|p_t^{(n)} - p_{t-1}^{(n)}\|_1^2 \hspace{3cm}(\text{using Lemma D.3})$$

$$\leq \mathcal{O}(\lambda) + \frac{\mathrm{KL}(u, w_1^{(n)})}{\eta_m} + 2\eta_m N\sum_{t=2}^{T}\sum_{n=1}^{N}\|p_t^{(n)} - p_{t-1}^{(n)}\|_1^2$$
$$+ \lambda\sum_{t=2}^{T-1}\sum_{k=1}^{d+2}u_k\|\widetilde{p}_t^{(n),k} - \widetilde{p}_{t-1}^{(n),k}\|_1^2 - \frac{\lambda}{4}\sum_{t=2}^{T}\|p_t^{(n)} - p_{t-1}^{(n)}\|_1^2, \tag{16}$$

where the last inequality uses the condition that $\lambda \leq \frac{1}{4\eta_m}$. $\qquad\square$

## D.3 Main Proofs in Section 6

In this section, we provide the proof for Theorem 6.4. Before showing the proof, we first provide an outline to highlight the technical novelties in proving Theorem 6.4.

### D.3.1 Proof Outline

As shown in previous literature (e.g. Anagnostides et al. [2022b], Zhang et al. [2022]), in order to show fast convergence, the key is to control the stability of the strategies between consecutive rounds. Anagnostides et al. [2022b] use log-barrier regularized online mirror descent to control the sum of the squared path-length between consecutive rounds over the horizon and all the players. However, due to the use of log-barrier regularizer, the obtained bound $\mathcal{O}(Nd^3 \log T)$ suffers from a larger

polynomial dependency of $d$ and $\log T$. Somewhat surprisingly, we show in the following theorem that if the game satisfies Definition 6.3, Algorithm 4 (with entropy regularizer) achieves a tighter $\mathcal{O}(N \log d)$ bound.

**Theorem D.7.** *If each player $n \in [N]$ applies Algorithm 4 with $\eta_m = \frac{1}{64N}$ and $\lambda = N$, then we have*

$$\sum_{t=2}^{T} \sum_{n=1}^{N} \|p_t^{(n)} - p_{t-1}^{(n)}\|_1^2 \leq \mathcal{O}(N \log d).$$

We provide a proof sketch for Theorem D.7 (with full proof deferred to Appendix D.3.2) and see why our modifications to both the meta learner and the base earners are crucial to achieve this. To prove Theorem D.7, we consider the each player $n$'s external regret, which can be decomposed as the meta learner regret with respect to $\mathcal{B}_{d+2}$ plus $\mathcal{B}_{d+2}$'s external regret:

$$\text{Reg}_n^{\text{Ext}}(u) = \underbrace{\sum_{t=1}^{T} \left\langle w_t^{(n)} - e_{d+2}, \ell_t^{(n),w} \right\rangle}_{\text{META-REGRET}} + \underbrace{\sum_{t=1}^{T} \left\langle \phi_t^{(n),d+2} - \mathbf{1}u^\top, p_t^{(n)} \ell_t^{(n)\top} \right\rangle}_{\text{BASE-REGRET}}.$$

Applying Lemma D.1, Lemma D.6, and some direct calculations, we can show that META-REGRET and BASE-REGRET are bounded as follows:

$$\text{META-REGRET} \leq \mathcal{O}(\lambda) + \frac{\text{KL}(e_{d+2}, w_1^{(n)})}{\eta_m} + 2\eta_m N \sum_{t=2}^{T} \sum_{n=1}^{N} \|p_t^{(n)} - p_{t-1}^{(n)}\|_1^2$$

$$+ \lambda \sum_{t=2}^{T-1} \|\widetilde{p}_t^{(n),d+2} - \widetilde{p}_{t-1}^{(n),d+2}\|_1^2 - \frac{\lambda}{4} \sum_{t=2}^{T} \|p_t^{(n)} - p_{t-1}^{(n)}\|_1^2 \tag{17}$$

$$\text{BASE-REGRET} \leq \frac{\log d}{\eta} + \eta \sum_{t=2}^{T} \|\ell_t^{(n)} - \ell_{t-1}^{(n)}\|_\infty^2 - \frac{1}{8\eta} \sum_{t=2}^{T} \|\widetilde{p}_t^{(n),d+2} - \widetilde{p}_{t-1}^{(n),d+2}\|_1^2, \tag{18}$$

where Eq. (18) uses the fact that $\phi_t^{(n),d+2} = \mathbf{1}\widetilde{p}_t^{(n),d+2\top}$. According to Lemma D.2, we can further upper bound both $\|\ell_t^{(n)} - \ell_{t-1}^{(n)}\|_\infty^2$ by $\mathcal{O}(N \sum_{n=1}^{N} \|p_t^{(n)} - p_{t-1}^{(n)}\|_1^2)$. Now, we see the importance of including correction terms in the meta-algorithm. Without $c_t^{(n)}$, the two negative term in Eq. (18) is *not enough* to cancel the above positive term. Thanks to the correction term, we are able to cancel the positive term $\mathcal{O}((\eta_m + \eta)N \sum_{t=2}^{T} \sum_{n=1}^{N} \|p_t^{(n)} - p_{t-1}^{(n)}\|_1^2)$ by using half of the negative term $-\frac{\lambda}{8} \sum_{t=1}^{T} \|p_t^{(n)} - p_{t-1}^{(n)}\|_1^2$, taking a summation over $n \in [N]$, and picking $\lambda$, $\eta_m$, and $\eta$ appropriately. Moreover, the positive term induced by the correction can be canceled by the negative term in Eq. (18). Therefore, summing over BASE-REGRET and META-REGRET for all $n \in [N]$ with $\eta_m = \Theta(1/N)$, $\eta = \Theta(1/N)$, and $\lambda = N$, we can obtain that $\sum_{n=1}^{N} \text{Reg}_n^{\text{Ext}} \leq \mathcal{O}(N^2 \log d) - \Omega(N \sum_{n=1}^{N} \sum_{t=2}^{T} \|p_t^{(n)} - p_{t-1}^{(n)}\|_1^2)$. Further using the property that $\sum_{n=1}^{N} \text{Reg}_n^{\text{Ext}} \geq 0$ finish the proof. □

Note that the above proof sketch indeed also proves an $\mathcal{O}(N \log d)$ external regret for each individual player. To obtain comparator-adaptive $\Phi$-regret, we first consider $\phi \in \Phi_b$ and obtain $\mathcal{O}(c_\phi \log d + N^2 \log d)$ by picking the meta learner's comparator $u \in \Delta(d+2)$ to be a distribution based on $\phi$. Then, the final result is achieved by taking a convex combination of the bound.

### D.3.2 Proof of Theorem D.7

In this section, we provide a detailed proof for Theorem D.7.

*Proof of Theorem D.7.* Fix $n \in [N]$ and consider the base-regret and the meta-regret for agent $n$ with respect to the base algorithm $\mathcal{A}_{d+2}$, which is Algorithm 9 handling the external regret. According to the construction of $\phi_t^{(n),d+2}$, we have $\phi_{t,i:}^{(n),d+2} = \widetilde{p}_t^{(n),d+2}$ for all $i \in [d]$, meaning that $\widetilde{p}_t^{(n),d+2}$ equals to the decision made by $\mathcal{A}_{d+2}$ at round $t$. Therefore, using Lemma D.1, the base regret of

$\mathcal{A}_{d+2}$ with respect to $u \in \Delta(d)$ is bounded as follows:

$$\sum_{t=1}^{T} \left\langle \widetilde{p}_{t,1}^{(n)} - u, \ell_t^{(n)} \right\rangle \leq \frac{\log d}{\eta} + \eta \sum_{t=2}^{T} \|\ell_t^{(n)} - \ell_{t-1}^{(n)}\|_\infty^2 - \frac{1}{8\eta} \sum_{t=2}^{T} \|\widetilde{p}_{t,1}^{(n)} - \widetilde{p}_{t-1,1}^{(n)}\|_1^2$$

$$\leq \frac{\log d}{\eta} + \eta(N-1) \sum_{t=2}^{T} \sum_{j \neq n} \|p_t^{(j)} - p_{t-1}^{(j)}\|_1^2 - \frac{1}{8\eta} \sum_{t=2}^{T} \|\widetilde{p}_{t,1}^{(n)} - \widetilde{p}_{t-1,1}^{(n)}\|_1^2. \tag{19}$$

As for meta-regret, applying Lemma D.6 with $u = e_{d+2}$ and noticing that $w_{1,d+2}^{(n)} = \frac{1}{4}$, we have

$$\sum_{t=1}^{T} \left\langle w_t^{(n)} - e_{d+2}, \ell_t^{(n),w} \right\rangle \leq \mathcal{O}(\lambda) + \frac{\log 4}{\eta_m} + 2\eta_m N \sum_{t=2}^{T} \sum_{n=1}^{N} \|p_t^{(n)} - p_{t-1}^{(n)}\|_1^2$$

$$+ \lambda \sum_{t=2}^{T-1} \|\widetilde{p}_{t,1}^{(n)} - \widetilde{p}_{t-1,1}^{(n)}\|_1^2 - \frac{\lambda}{4} \sum_{t=2}^{T} \|p_t^{(n)} - p_{t-1}^{(n)}\|_1^2. \tag{20}$$

Summing up Eq. (19) and Eq. (20), we can bound the external regret for player $n$ as follows:

$$\text{Reg}_n^{\text{Ext}} = \sum_{t=1}^{T} \left\langle p_t^{(n)} - u, \ell_t^{(n)} \right\rangle$$

$$\leq \mathcal{O}(\lambda) + \frac{\log 4}{\eta_m} + \frac{\log d}{\eta} + (2\eta_m + \eta) N \sum_{t=2}^{T} \sum_{n=1}^{N} \|p_t^{(n)} - p_{t-1}^{(n)}\|_1^2$$

$$+ \lambda \sum_{t=2}^{T-1} \|\widetilde{p}_t^{(n),d+2} - \widetilde{p}_{t-1}^{(n),d+2}\|_1^2 - \frac{1}{8\eta} \sum_{t=2}^{T} \|\widetilde{p}_t^{(n),d+2} - \widetilde{p}_{t-1}^{(n),d+2}\|_1^2 - \frac{\lambda}{4} \sum_{t=2}^{T} \|p_t^{(n)} - p_{t-1}^{(n)}\|_1^2$$

$$\leq \mathcal{O}(\lambda) + \frac{\log 4}{\eta_m} + \frac{\log d}{\eta} + (2\eta_m + \eta) N \sum_{t=2}^{T} \sum_{n=1}^{N} \|p_t^{(n)} - p_{t-1}^{(n)}\|_1^2 - \frac{\lambda}{4} \sum_{t=2}^{T} \|p_t^{(n)} - p_{t-1}^{(n)}\|_1^2, \tag{21}$$

where the last inequality uses $\lambda = N \leq \frac{1}{8\eta} = 2N$. Taking summation over $n \in [N]$ and using Definition 6.3 that $\sum_{n=1}^{N} \text{Reg}_n^{\text{Ext}} \geq 0$, we know that

$$0 \leq \sum_{n=1}^{N} \text{Reg}_n^{\text{Ext}}$$

$$\leq \mathcal{O}(N\lambda) + \frac{N \log 4}{\eta_m} + \frac{N \log d}{\eta} + \left((2\eta_m + \eta)N^2 - \frac{\lambda}{4}\right) \sum_{t=2}^{T} \sum_{n=1}^{N} \|p_t^{(n)} - p_{t-1}^{(n)}\|_1^2.$$

According to the choice of $\lambda$, we know that $\frac{\lambda}{8} = \frac{N}{8} \geq \frac{3N}{32} = (2\eta_m + \eta)N^2$. Rearranging the terms gives

$$\frac{N}{8} \sum_{t=2}^{T} \sum_{n=1}^{N} \|p_t^{(n)} - p_{t-1}^{(n)}\|_1^2 \leq \mathcal{O}(N^2 \log d),$$

which finishes the proof. $\qquad\square$

### D.3.3 Proof of Theorem 6.4

Now we prove our main results Theorem 6.4 for multi-agent games. Specifically, we split the proof into three parts and first prove the external regret guarantee.

**Theorem D.8.** *Suppose that all agents run Algorithm 4 with $\lambda = N$, $\eta_m = \frac{1}{64N}$. Then, we have $\text{Reg}_n^{\text{Ext}} \leq \mathcal{O}(N \log d)$.*

*Proof.* According to Eq. (21), we know that

$$\text{Reg}_n \leq \mathcal{O}(\lambda) + \frac{\log 4}{\eta_m} + \frac{\log d}{\eta} + (2\eta_m + \eta) N \sum_{t=2}^{T} \sum_{n=1}^{N} \|p_t^{(n)} - p_{t-1}^{(n)}\|_1^2$$

$$\leq \mathcal{O}(N + N \log d) + \mathcal{O}\left( \sum_{t=2}^{T} \sum_{n=1}^{N} \|p_t^{(n)} - p_{t-1}^{(n)}\|_1^2 \right)$$

$$\leq \mathcal{O}(N \log d),$$

where the second inequality is due to the choice of $\eta_m, \eta$, and the final inequality is due to Theorem D.7. $\qquad \square$

Next, we prove our results for $\Phi$-regret. As we sketched in Appendix D.3.1, we first prove our results for binary transformation matrices $\phi \in \Phi_b$. First, the following theorem shows that our algorithm achieves $\text{Reg}_n(\phi) = \mathcal{O}(N(d - d_\phi^{\text{self}}) \log d + N^2 \log d)$ for all $\phi \in \Phi_b$.

**Theorem D.9.** *Suppose that all agents run Algorithm 4 with $\lambda = N$, $\eta_m = \frac{1}{64N}$. Then, we have $\text{Reg}_n(\phi) \leq \mathcal{O}((d - d_\phi^{\text{self}})N \log d + N^2 \log d)$ for all $\phi \in \Phi_b$.*

*Proof.* To achieve $\text{Reg}_n(\phi) \leq \mathcal{O}(N(d - d_\phi^{\text{self}}) \log d + N^2 \log d)$, we consider the regret with respect to base algorithm $\mathcal{A}_{d+1}$. According to Lemma D.6 and Lemma D.4, we bound the meta-regret as follows:

$$\sum_{t=1}^{T} \left\langle w_t^{(n)} - e_{d+1}, \ell_t^{(n),w} \right\rangle$$

$$\leq \mathcal{O}(\lambda) + \frac{\log 4}{\eta_m} + 2\eta_m N \sum_{t=2}^{T} \sum_{n=1}^{N} \|p_t^{(n)} - p_{t-1}^{(n)}\|_1^2 + \lambda \sum_{t=2}^{T-1} \|\widetilde{p}_{t,i}^{(n)} - \widetilde{p}_{t-1,i}^{(n)}\|_1^2 - \frac{\lambda}{4} \sum_{t=1}^{T} \|p_t^{(n)} - p_{t-1}^{(n)}\|_1^2$$

$$\leq \mathcal{O}(\lambda) + \frac{\log 4}{\eta_m} + 2\eta_m N \sum_{t=2}^{T} \sum_{n=1}^{N} \|p_t^{(n)} - p_{t-1}^{(n)}\|_1^2 + 2\lambda \sum_{t=2}^{T-1} \|p_t^{(n)} - p_{t-1}^{(n)}\|_1^2$$

$$+ 2\lambda \sum_{j=1}^{d} \sum_{t=2}^{T-1} \|\phi_{t,j:}^{(n),d+1} - \phi_{t-1,j:}^{(n),d+1}\|_1^2,$$

where the second inequality uses Lemma D.4. According to the analysis similar to Theorem C.1, we know that base-regret of $\mathcal{A}_{d+1}$ can be bounded as follows:

$$\sum_{t=1}^{T} \left\langle \phi_t^{(n),d+1} - \phi, p_t^{(n)} \ell_t^{(n)\top} \right\rangle$$

$$= \sum_{t=1}^{T} \sum_{i=1}^{d} \left\langle \phi_{t,i:}^{(n),d+2} - \phi(e_i), p_{t,i}^{(n)} \cdot \ell_t^{(n)} \right\rangle$$

$$\leq \sum_{i=1}^{d} \left( \frac{\text{KL}(\phi(e_i), \psi_{i:}^{d+1})}{\eta} + \eta \sum_{t=1}^{T} \left\| p_{t,i}^{(n)} \cdot \ell_t^{(n)} - p_{t-1,i}^{(n)} \cdot \ell_{t-1}^{(n)} \right\|_\infty^2 - \frac{1}{8\eta} \sum_{t=1}^{T} \left\| \phi_{t,i:}^{(n),d+1} - \phi_{t-1,i:}^{(n),d+1} \right\|_1^2 \right)$$

$$\hfill \text{(using Lemma D.1)}$$

$$= \frac{\log \frac{1}{\pi_{\psi^{d+1}}(\phi)}}{\eta} - \frac{1}{8\eta} \sum_{i=1}^{d} \sum_{t=1}^{T} \left\| \phi_{t,i:}^{(n),d+1} - \phi_{t-1,i:}^{(n),d+1} \right\|_1^2$$

$$+ \eta \sum_{i=1}^{d} \sum_{t=1}^{T} \left\| p_{t,i}^{(n)} \cdot \ell_t^{(n)} - p_{t-1,i}^{(n)} \cdot \ell_{t-1}^{(n)} \right\|_\infty^2$$

$$\leq \frac{2(d - d_\phi^{\text{self}}) \log d + 1}{\eta} - \frac{1}{8\eta} \sum_{t=1}^{T} \sum_{i=1}^{d} \left\| \phi_{t,i:}^{(n),d+1} - \phi_{t-1,i:}^{(n),d+1} \right\|_1^2 \hfill \text{(according to Eq. (4))}$$

$$+ 2\eta \sum_{t=1}^{T} \sum_{i=1}^{d} \left( \left\| p_{t,i}^{(n)} \cdot \ell_t^{(n)} - p_{t,i}^{(n)} \cdot \ell_{t-1}^{(n)} \right\|_\infty^2 + \left\| p_{t,i}^{(n)} \cdot \ell_{t-1}^{(n)} - p_{t-1,i}^{(n)} \cdot \ell_{t-1}^{(n)} \right\|_\infty^2 \right)$$

$$\leq \frac{2(d - d_\phi^{\text{self}}) \log d + 1}{\eta} - \frac{1}{8\eta} \sum_{t=1}^{T} \sum_{i=1}^{d} \left\| \phi_{t,i:}^{(n),d+1} - \phi_{t-1,i:}^{(n),d+1} \right\|_1^2$$

$$+ 2\eta \sum_{t=1}^{T} \sum_{i=1}^{d} \left( p_{t,i}^{(n)^2} \left\| \ell_t^{(n)} - \ell_{t-1}^{(n)} \right\|_\infty^2 + \left| p_{t,i}^{(n)} - p_{t-1,i}^{(n)} \right|^2 \right)$$

$$\leq \frac{2(d - d_\phi^{\text{self}}) \log d + 1}{\eta} - \frac{1}{8\eta} \sum_{t=1}^{T} \sum_{i=1}^{d} \left\| \phi_{t,i:}^{(n),d+1} - \phi_{t-1,i:}^{(n),d+1} \right\|_1^2$$

$$+ 2\eta \sum_{t=1}^{T} \left( \left\| \ell_t^{(n)} - \ell_{t-1}^{(n)} \right\|_\infty^2 + \left\| p_t^{(n)} - p_{t-1}^{(n)} \right\|_1^2 \right)$$

$$\leq \frac{2(d - d_\phi^{\text{self}}) \log d + 1}{\eta} - \frac{1}{8\eta} \sum_{t=1}^{T} \sum_{i=1}^{d} \left\| \phi_{t,i:}^{(n),d+1} - \phi_{t-1,i:}^{(n),d+1} \right\|_1^2$$

$$+ 2\eta(N-1) \sum_{t=2}^{T} \sum_{j \neq n} \left\| p_t^{(j)} - p_{t-1}^{(j)} \right\|_1^2 + 2\eta \sum_{t=1}^{T} \left\| p_t^{(n)} - p_{t-1}^{(n)} \right\|_1^2. \qquad \text{(using Lemma D.2)}$$

Summing up the base-regret and meta-regret, we can obtain that

$$\sum_{t=1}^{T} \left\langle \phi_t - \phi, p_t^{(n)} \ell_t^{(n)\top} \right\rangle$$

$$\leq \mathcal{O}(\lambda) + \frac{\log 4}{\eta_m} + 2\eta_m N \sum_{t=2}^{T} \sum_{n=1}^{N} \|p_t^{(n)} - p_{t-1}^{(n)}\|_1^2 + 2\lambda \sum_{t=2}^{T-1} \|p_t^{(n)} - p_{t-1}^{(n)}\|_1^2$$

$$+ 2\lambda \sum_{j=1}^{d} \sum_{t=2}^{T-1} \|\phi_{t,j:}^{(n),d+1} - \phi_{t-1,j:}^{(n),d+1}\|_1^2$$

$$+ \frac{2(d - d_\phi^{\text{self}}) \log d + 1}{\eta} - \frac{1}{8\eta} \sum_{t=2}^{T} \sum_{i=1}^{d} \left\| \phi_{t,i:}^{(n),d+1} - \phi_{t-1,i:}^{(n),d+1} \right\|_1^2$$

$$+ 2\eta(N-1) \sum_{t=2}^{T} \sum_{j \neq n} \left\| p_t^{(j)} - p_{t-1}^{(j)} \right\|_1^2 + 2\eta \sum_{t=2}^{T} \left\| p_t^{(n)} - p_{t-1}^{(n)} \right\|_1^2$$

$$\leq \mathcal{O}(\lambda) + \frac{\log 4}{\eta_m} + \frac{2(d - d_\phi^{\text{self}}) \log d + 1}{\eta}$$

$$+ (2\eta_m N + 2\eta N + \lambda) \sum_{t=2}^{T} \sum_{j=1}^{N} \|p_t^{(j)} - p_{t-1}^{(j)}\|_1^2. \qquad \text{(since } 2\lambda = 2N \leq \frac{1}{8\eta})$$

Since $\lambda = N$, $\eta_m = \frac{1}{64N}$ and $\eta = \frac{1}{16N}$ and using Theorem D.7, we know that

$$\sum_{t=1}^{T} \left\langle \phi_t - \phi, p_t^{(n)} \ell_t^{(n)\top} \right\rangle \leq \mathcal{O} \left( N(d - d_\phi^{\text{self}}) \log d + N^2 \log d \right).$$

$\square$

Next, we prove our second bound with respect to $d - d_\phi^{\text{unif}} + 1$.

**Theorem D.10.** *Suppose that all agents run Algorithm 4 with $\lambda = N$, $\eta_m = \frac{1}{64N}$. Then, we have $\text{Reg}_n(\phi) \leq \mathcal{O}((d - d_\phi^{\text{unif}} + 1)N \log d + N^2 \log d)$ for all $\phi \in \Phi_b$.*

*Proof.* Given $\phi \in \Phi_b$, suppose that the most frequent element in $\{\phi(e_1), \ldots, \phi(e_d)\}$ is $e_{i_0}$ for some $i_0 \in [d]$. According to the definition of $d_\phi^{\text{unif}}$, we know that there exists $d_\phi^{\text{unif}}$ number of $i \in [d]$ such that $\phi(e_i) = e_{i_0}$. To bound $\text{Reg}_n(\phi)$, we compare to the base-learner $\mathcal{A}_{i_0}$. Applying Lemma D.6 gives us

$$\sum_{t=1}^{T} \left\langle w_t^{(n)} - e_{i_0}, \ell_t^{(n),w} \right\rangle$$

$$\leq \mathcal{O}(\lambda) + \frac{\log 4d}{\eta_m} + 2\eta_m N \sum_{t=2}^{T} \sum_{n=1}^{N} \|p_t^{(n)} - p_{t-1}^{(n)}\|_1^2 + \lambda \sum_{t=2}^{T-1} \|\widetilde{p}_t^{(n),i_0} - \widetilde{p}_{t-1}^{(n),i_0}\|_1^2$$

$$- \frac{\lambda}{4} \sum_{t=1}^{T} \|p_t^{(n)} - p_{t-1}^{(n)}\|_1^2$$

$$\leq \mathcal{O}(\lambda) + \frac{\log 4d}{\eta_m} + 2\eta_m N \sum_{t=2}^{T} \sum_{n=1}^{N} \|p_t^{(n)} - p_{t-1}^{(n)}\|_1^2 + 2\lambda \sum_{t=2}^{T-1} \|p_t^{(n)} - p_{t-1}^{(n)}\|_1^2$$

$$+ 2\lambda \sum_{j=1}^{d} \sum_{t=2}^{T-1} \|\phi_{t,j:}^{(n),i_0} - \phi_{t-1,j:}^{(n),i_0}\|_1^2,$$

where the second inequality is because Lemma D.4. Now we analyze the base-algorithm performance of $\text{Alg}_{i_0}$ against $\phi$:

$$\sum_{t=1}^{T} \left\langle \phi_t^{(n),i_0} - \phi, p_t^{(n)} \ell_t^{(n)\top} \right\rangle$$

$$= \sum_{t=1}^{T} \sum_{i=1}^{d} \left\langle \phi_{t,i:}^{(n),i_0} - \phi(e_i), p_{t,i}^{(n)} \cdot \ell_t^{(n)} \right\rangle$$

$$\leq \sum_{i=1}^{d} \left( \frac{\text{KL}(\phi(e_i), \psi_{i:}^{i_0})}{\eta} + \eta \sum_{t=1}^{T} \left\| p_{t,i}^{(n)} \cdot \ell_t^{(n)} - p_{t-1,i}^{(n)} \cdot \ell_{t-1}^{(n)} \right\|_\infty^2 - \frac{1}{8\eta} \sum_{t=1}^{T} \left\| \phi_{t,i:}^{(n),i_0} - \phi_{t-1,i:}^{(n),i_0} \right\|_1^2 \right)$$

$$\leq \frac{\log \frac{1}{\pi_{\psi^{i_0}}(\phi)}}{\eta} - \frac{1}{8\eta} \sum_{t=1}^{T} \sum_{i=1}^{d} \left\| \phi_{t,i:}^{(n),i_0} - \phi_{t-1,i:}^{(n),i_0} \right\|_1^2 + 2\eta \sum_{t=1}^{T} \left( \left\| \ell_t^{(n)} - \ell_{t-1}^{(n)} \right\|_\infty^2 + \left\| p_t^{(n)} - p_{t-1}^{(n)} \right\|_1^2 \right)$$

$$\leq \frac{2(d - d_\phi^{\text{unif}}) \log d + 1}{\eta} - \frac{1}{8\eta} \sum_{t=1}^{T} \sum_{i=1}^{d} \left\| \phi_{t,i:}^{(n),i_0} - \phi_{t-1,i:}^{(n),i_0} \right\|_1^2 \qquad \text{(according to Eq. (5))}$$

$$+ 2\eta \sum_{t=1}^{T} \left( \left\| \ell_t^{(n)} - \ell_{t-1}^{(n)} \right\|_\infty^2 + \left\| p_t^{(n)} - p_{t-1}^{(n)} \right\|_1^2 \right)$$

$$\leq \frac{2(d - d_\phi^{\text{unif}}) \log d + 1}{\eta} - \frac{1}{8\eta} \sum_{t=1}^{T} \sum_{i=1}^{d} \left\| \phi_{t,i:}^{(n),i_0} - \phi_{t-1,i:}^{(n),i_0} \right\|_1^2$$

$$+ 2\eta(N-1) \sum_{t=2}^{T} \sum_{i \neq n} \left\| p_t^{(i)} - p_{t-1}^{(i)} \right\|_1^2 + 2\eta \sum_{t=1}^{T} \left\| p_t^{(n)} - p_{t-1}^{(n)} \right\|_1^2. \qquad \text{(using Lemma D.2)}$$

Summing up the meta-regret and the base-regret, we can obtain that

$$\text{Reg}_n(\phi) = \sum_{t=1}^{T} \left\langle w_t^{(n)} - u, \ell_t^{(n),w} \right\rangle + \sum_{t=1}^{T} \left\langle \phi_t^{(n),i_0} - \phi, p_t^{(n)} \ell_t^{(n)\top} \right\rangle$$

$$\leq \mathcal{O}(\lambda) + \frac{\log 4d}{\eta_m} + 2\eta_m N \sum_{t=2}^{T} \sum_{n=1}^{N} \|p_t^{(n)} - p_{t-1}^{(n)}\|_1^2 + 2\lambda \sum_{t=2}^{T-1} \|p_t^{(n)} - p_{t-1}^{(n)}\|_1^2$$

$$+ 2\lambda \sum_{i=1}^{d} \sum_{t=2}^{T-1} \|\phi_{t,i:}^{(n),i_0} - \phi_{t-1,i:}^{(n),i_0}\|_1^2$$

$$+ \frac{2(d - d_\phi^{\text{unif}})\log d + 1}{\eta} - \frac{1}{8\eta} \sum_{t=2}^{T} \sum_{i=1}^{d} \left\| \phi_{t,i:}^{(n),i_0} - \phi_{t-1,i:}^{(n),i_0} \right\|_1^2$$

$$+ 2\eta(N-1) \sum_{t=2}^{T} \sum_{j \neq n} \left\| p_t^{(j)} - p_{t-1}^{(j)} \right\|_1^2 + 2\eta \sum_{t=2}^{T} \left\| p_t^{(n)} - p_{t-1}^{(n)} \right\|_1^2$$

$$= \mathcal{O}(\lambda) + \frac{\log 4}{\eta_m} + \frac{2(d - d_\phi^{\text{unif}})\log d + 1}{\eta}$$

$$+ (2\eta_m N + 2\eta N + \lambda) \sum_{t=2}^{T} \sum_{j=1}^{N} \| p_t^{(j)} - p_{t-1}^{(j)} \|_1^2. \qquad \text{(since } 2\lambda = 2N \leq \frac{1}{8\eta})$$

$$\leq \mathcal{O}\left( N(d - d_\phi^{\text{unif}} + 1)\log d + N^2 \log d \right),$$

where the last inequality is by picking $\eta = \frac{1}{16N}$ and using Theorem D.7. $\qquad\square$

Finally, we are ready to prove Theorem 6.4 by combining Theorem D.9 and Theorem D.10.

*Proof of Theorem 6.4.* Combining Theorem D.9 and Theorem D.10, we know that for any $\phi \in \Phi_b$,

$$\text{Reg}_n(\phi) \leq (c_\phi N \log d + N^2 \log d).$$

Then, for $\phi \in \mathcal{S}$, define $q_\phi = \text{argmin}_{q \in Q_\phi} \mathbb{E}_{\phi' \sim q}[c_{\phi'}]$. Then, we know that $c_\phi = \mathbb{E}_{\phi' \sim q_\phi}[c_{\phi'}]$ and

$$\text{Reg}_n(\phi) = \mathbb{E}_{\phi' \sim q_\phi}[\text{Reg}_n(\phi')] \leq \mathcal{O}\left( \mathbb{E}_{\phi' \sim q_\phi}[c_{\phi'}] N \log d + N^2 \log d \right) \leq \mathcal{O}(c_\phi N \log d + N^2 \log d).$$

Combining the above with Theorem D.8 finishes the proof. $\qquad\square$

