# OpenReview forum: "Comparator-Adaptive $\Phi$-Regret: Improved Bounds, Simpler Algorithms, and Applications to Games"
_NeurIPS.cc/2025/Conference — NeurIPS 2025 spotlight_

### Official Review · Reviewer_zKH9 · 2025-06-27

**Clarity:** 4
**Significance:** 3
**Originality:** 4
**Rating:** 5
**Confidence:** 4

**Summary:**

The paper considers minimizing the $\Phi$ regret in the expert problem, and propose new algorithms achieving certain regret bounds that adapt to a sparsity based complexity measure of each comparing $\phi$ instance, improving the results from a recent work (Lu et al., 2025). Different from the existing algorithm using wavelet inspired matrix features, the proposed algorithms are based on the arguably simpler idea of expert aggregation, with the twists being the incorporation of a carefully designed prior distribution over all possible $\phi$s, as well as the idea of kernelization from (Farina et al., 2022). Finally, a variant of the algorithm is proposed to achieve an adaptive version of "fast rate" for learning in games, improving (Anagnostides et al., 2022).

**Questions:**

In the fast kernelized algorithm (Algorithm 6), I wonder if the computation of $\phi_t$ can be further reduced to matrix multiplication, thus $d^\omega$ time.

**Ethical Concerns:**

["NO or VERY MINOR ethics concerns only"]

**Limitations:**

Yes

**Paper Formatting Concerns:**

I do not see any formatting concerns.

**Quality:**

4

**Strengths And Weaknesses:**

Overall, I think the paper has several notable strengths making it well above the acceptance threshold.

1. The results are strong as it removes several suboptimalities from the prior work and tightens its complexity description for the vast majority of transformation rules that are non-binary.

2. The ideas behind the algorithms are substantially different from that of Lu et al, bringing a novel, complementary perspective to the problem. In general, adaptivity in online learning is typically achieved by either aggregating different nonadaptive subroutines or the "parameter-free" type of approaches. Their connections and differences have provided interesting insights to the field, and the present work contributes to this thread as well.

3. The application to the fast rates in games is a nontrivial addition.

4. Writing is exceptionally clear despite the complicated technicalities.

On the flip side, I would suggest adding an in-depth discussion on the computational complexity. I understand the key motivation of this adaptive $\Phi$-regret problem as providing a beyond-the-worst improvement over the classical Blum-Mansour algorithm, therefore BM could be seen as the natural baseline. There, given the fixed point computational oracle which is matrix multiplication time ($d^3$ in practice but $d^\omega$ in theory), the essential online learning occurs in a $d^2$ dimensional space, therefore the fixed point oracle is the only computational bottleneck and there has been some attempts accelerating it (such as power method, discussed in Appendix I.3 by Zhang et al. 2024). (Lu et al, 2025) is computationally similar, but in the present work the essential online learning appears to take $d^3\log d$ time per iteration for both of the two proposed algorithms, dominating the fixed point computation. This makes the comparison to BM and Lu et al a bit more subtle, and some clarification would be nice.

Zhang, Brian, et al. "Efficient $\Phi $-Regret Minimization with Low-Degree Swap Deviations in Extensive-Form Games." Advances in Neural Information Processing Systems 37 (2024): 125192-125230.

---

> ### Author Rebuttal · Authors · 2025-07-30
>
> >Q1: In the fast kernelized algorithm (Algorithm 6), I wonder if the computation of $\phi_t$ can be further reduced to matrix multiplication, thus $d^\omega$ time.
>
> This is an excellent question. In fact, we now realize that each $\\phi\_t$ can be computed in $\\mathcal{O}(d^2)$ time (so better than the $\\mathcal{O}(d^3)$ time we provided in Algorithm 6 and also better than what the reviewer suspected, $\\mathcal{O}(d^{\\omega})$).
> This makes the computational complexity of our algorithm exactly the same as Blum-Mansour and Lu et al. [2025].
> We provide the proof below and will update Algorithm 6 this way in the next version.
> Thanks again for your comment.
>
> First, note that using the definition of $\\phi\_{t+1}$ in Algorithm 6 and the definition of $\\psi^k$, one can verify that the entry of $\\phi\_{t+1}$ takes the following form: if $i\neq j$,
> \\[
> \(\phi_{t+1})\_{ij}=
> \\begin{aligned}
> \\frac{d-1}{c\_{t}d(d-2)}(V\_t)\_{ij}\\left(\\frac{d-2}{d-1}(C\_t)\_{ij}+\\frac{1}{d(d-1)}(S\_t)\_i+\\frac{1}{d-1}(C\_t)\_{i,d+1}\\right),\\end{aligned}
> \\]
>
> if $i=j$,
> \\[
> \\begin{aligned}
> \(\phi_{t+1})\_{ij}=
> \\frac{d-1}{c\_{t}d(d-2)}(V\_t)\_{ij}\\left(\\frac{d-2}{d-1}(C\_t)\_{ij}+\\frac{1}{d(d-1)}(S\_t)\_i+(d-1)(C\_t)\_{i,d+1}\\right),
> \\end{aligned}
> \\]
>
> where
>
>  - $V_t\\in\\mathbb{R}^{d\\times d}$ is defined by $(V\_t)\_{ik} = \\frac{\\exp(-\\eta(L\_t)\_{ik})}{\\sum\_{j=1}^d \\exp(-\\eta(L\_t)\_{ij})}$ and can be calculated in $\\mathcal{O}(d^2)$ time.
>  - $c_{t}\\in \\mathbb{R}$ is defined as
> $\\frac{1}{d} \\sum\_{k=1}^d\\prod\_{i=1}^d\\left((V\_t)\_{ik}+\\frac{1}{d(d-2)}\\Big) + \\prod\_{i=1}^d\\Big((V\_t)\_{ii}+\\frac{1}{d(d-2)}\\right)$. With $V\_t$, $c\_{t}$ can be computed in $\\mathcal{O}(d^2)$ time.
>  - $C_t\\in\\mathbb{R}^{d\\times (d+1)}$ is defined as follows: for all $k\\in[d]$, $(C\_t)\_{ik} = \\frac{(v\_{t})\_k}{(V\_t)\_{ik}+\\frac{1}{d(d-2)}}$ and $(C\_t)\_{i,d+1} = \\frac{u\_t}{(V\_t)\_{ii}+\\frac{1}{d(d-2)}}$
> where $(v\_{t})\_k = \\prod_{i=1}^d((V\_t)\_{ik}+\\frac{1}{d(d-2)})$
> and $u\_t = \\prod\_{i=1}^d((V\_t)\_{ii}+\\frac{1}{d(d-2)})$.
> Again, with $V\_t$, $v\_t$ and $u\_t$ can be computed in $\\mathcal{O}(d^2)$ time, and then $C\_t$ can be computed in $\\mathcal{O}(d^2)$ time too.
>  - $S\_t\\in \\mathbb{R}^d$ is defined by
> $(S\_t)\_i = \\sum\_{k=1}^d (C\_t)\_{ik}$ for $i\\in [d]$. With $C\_t$, $S\_t$ can be computed in $\\mathcal{O}(d^2)$ time too.
>
> Therefore, it is now clear that computing $\\phi\_{t+1}$ also only takes $\\mathcal{O}(d^2)$ time.

---

> > ### Comment · Reviewer_zKH9 · 2025-08-02
> >
> > Thank you for your rebuttal, this is a very nice result that perfectly addressed my comment. I'll strongly recommend this paper in the discussion phase.

---

### Official Review · Reviewer_FyQk · 2025-06-28

**Clarity:** 4
**Significance:** 3
**Originality:** 3
**Rating:** 5
**Confidence:** 3

**Summary:**

This paper investigates instance-dependent \Phi-regret bounds. Building on techniques from prior work, the authors propose a refined complexity measure for the comparator, develop two efficient algorithms, and derive improved regret bounds.

**Questions:**

See above

**Ethical Concerns:**

["NO or VERY MINOR ethics concerns only"]

**Final Justification:**

I have no outstanding concerns.

**Limitations:**

Yes

**Quality:**

3

**Strengths And Weaknesses:**

The paper is well written: the discussion of existing work is fair, the motivations behind the algorithmic development are clearly articulated, and the limitations are also acknowledged.

The construction of the prior is interesting, but the paper lacks a clear motivation for introducing it, and it is unclear whether this prior (and the resulting c_\phi) is the best possible choice. While the results presented are fairly comprehensive, the paper would benefit from a more thorough justification of the complexity measure c_\phi. It indeed interpolates between external and swap regret, but it is unclear what precise middle ground it captures. Moreover, c_\phi is defined as the minimum of two complexities, which seems somewhat ad hoc. It would strengthen the paper to provide a justification showing that c_\phi is inherent: for example, demonstrating that for some reasonable class of comparators (lying between \Phi-external/internal and \Phi-best), c_\phi is in fact the best complexity measure achievable.

---

> ### Author Rebuttal · Authors · 2025-07-30
>
> Thanks for your positive review and valuable comments. We address the issues mentioned in your review below.
>
> >Q1: The construction of the prior is interesting, but the paper lacks a clear motivation for introducing it, and it is unclear whether this prior (and the resulting c_\phi) is the best possible choice. While the results presented are fairly comprehensive, the paper would benefit from a more thorough justification of the complexity measure c_\phi. It indeed interpolates between external and swap regret, but it is unclear what precise middle ground it captures. Moreover, c_\phi is defined as the minimum of two complexities, which seems somewhat ad hoc. It would strengthen the paper to provide a justification showing that c_\phi is inherent: for example, demonstrating that for some reasonable class of comparators (lying between \Phi-external/internal and \Phi-best), c_\phi is in fact the best complexity measure achievable.
>
> While we also do not believe that our $c_\phi$ is the ``best'' complexity measure (in the sense that any reasonable algorithms must incur $\mathrm{Reg}(\phi)=\Omega(\sqrt{c_\phi T\log d})$ for all $\phi$), we note that we can at least say that
> for any integer $k \in [d]$ and any algorithm, there exists a $d$-expert problem and a $\phi$ with $c_\phi \leq k+1$, such that $\mathrm{Reg}(\phi)$ is at least $\Omega(\sqrt{c_\phi T\log c_\phi})$ for this algorithm.
>
> To see this, simply let $d-k$ experts be dummy experts who always have maximum loss $1$, while the other $k$ experts follow the swap regret lower bound construction of Ito [2020] scaled by $1/2$.
> We then define $\phi$ in this way.
> First, for a non-dummy expert, $\phi$ maps it to another non-dummy expert optimally (to minimize the total loss after swap).
> Then, to define $\phi$'s behavior for dummy experts, we consider two cases:
> in the case where the algorithm picks dummy experts for more than $\sqrt{kT\log k}$ times, we let $\phi$ map all dummy experts to a fixed non-dummy expert (in which case $d_\phi^{\text{unif}} \geq d-k$); otherwise, all dummy experts are mapped to themselves (in which case $d_\phi^{\text{self}} \geq d-k$).
> Note that in both cases, we have $c_\phi \leq k+1$.
> Moreover, in the first case, $\mathrm{Reg}(\phi)$ is at least $\frac{1}{2}\sqrt{kT\log k}$ since whenever the algorithm picks a dummy expert $i$, it gets loss $1$ while expert $\phi(i)$ gets loss at most $1/2$ (and the rest of the regret is non-negative by construction).
> In the second case, $\mathrm{Reg}(\phi)$ is essentially the swap regret of the algorithm for a $k$-expert problem that lasts for at least $T-\sqrt{kT\log k}$ rounds,
> and since the instance comes from the lower bound construction of Ito [2020], we also have $\mathrm{Reg}(\phi)=\Omega(\sqrt{k\log(k)(T-\sqrt{kT\log k})}) = \Omega(\sqrt{kT\log k})$.

---

> > ### Comment · Reviewer_FyQk · 2025-08-05
> >
> > Thank you for the response. It’s helpful to see the lower bound, though I was hoping for something more instance-specific. I maintain my positive evaluation of the paper and support its acceptance.

---

### Official Review · Reviewer_WnBk · 2025-06-30

**Clarity:** 2
**Significance:** 3
**Originality:** 3
**Rating:** 5
**Confidence:** 2

**Summary:**

This paper introduces a novel framework for comparator-adaptive $\Phi$-regret, significantly improving upon the work of Lu et al. (2025). The authors propose two efficient algorithms—based on kernelized Multiplicative Weight Update (MWU) and a prior-aware BM-reduction—to achieve tighter regret bounds, removing unnecessary terms from the prior work.

**Questions:**

- Given the focus on upper bounds for comparator-adaptive $\Phi$-regret, is it possible to provide a lower bound analysis to establish the optimality of the proposed bounds, especially in relation to the complexity measure $c\_{\phi}$?
- Regarding the kernelized MWU algorithm (Section 4), the paper mentions an $O(d^3)$ complexity per iteration. How does this complexity impact the algorithm’s scalability for large $d$? Are there practical optimizations considered to handle high-dimensional scenarios?
- Can the framework be extended to handle heterogeneous agents with different action spaces and loss functions, as encountered in real-world games, and if so, how would the current analysis need to be adapted to accommodate such heterogeneity?

**Ethical Concerns:**

["NO or VERY MINOR ethics concerns only"]

**Final Justification:**

The rebuttal has addressed my questions and improved upon the original results. As this is a solid theoretical paper that exceeds the acceptance threshold, I have decided to increase my score.

**Limitations:**

- Game Assumptions: The nonnegative-social-external-regret assumption restricts the applicability to specific game classes, limiting generality.
- Lack of Experiments: The absence of empirical results prevents validation of the algorithms’ efficiency and practical utility, especially compared to the method in Lu et al. (2025).

**Paper Formatting Concerns:**

No formatting issues were noticed.

**Quality:**

3

**Strengths And Weaknesses:**

**Strengths:**

- The paper achieves tighter regret bounds, matching the minimax-optimal bounds for standard regret notions (external, internal, swap).
- The proposed kernelized MWU and BM-reduction approaches are computationally more efficient than prior methods.
- The extension to games is notable, providing the first adaptive and accelerated $\Phi$-equilibrium guarantee.

**Weaknesses:**

- Although the work is purely theoretical, lacking experimental results to demonstrate practical performance. Empirical evaluation on real-world datasets or game instances would strengthen the claims.
- The $\Phi$-equilibrium bound includes an $O(N^2\log d)$ term (Theorem 6.4), which scales quadratically with the number of players $N$. This may limit applicability to large multi-agent systems.

---

> ### Author Rebuttal · Authors · 2025-07-30
>
> Thank you for your positive review and comments. We address your questions below.
>
> >Q1: Given the focus on upper bounds for comparator-adaptive $\Phi$-regret, is it possible to provide a lower bound analysis to establish the optimality of the proposed bounds, especially in relation to the complexity measure $c\_{\phi}$?
>
> We already mentioned in the paper that our bound is optimal for the special case of external/internal/swap regret.
> In fact, by a simple argument, one can establish the following stronger lower bound as well: for any integer $k \in [d]$ and any algorithm, there exists a $d$-expert problem and a $\phi$ with $c\_\phi \leq k+1$, such that $\mathrm{Reg}(\phi)$ is at least $\Omega(\sqrt{c\_\phi T\log c\_\phi})$ for this algorithm.
>
> To see this, simply let $d-k$ experts be dummy experts who always have maximum loss $1$, while the other $k$ experts follow the swap regret lower bound construction of Ito [2020] scaled by $1/2$.
> We then define $\phi$ in this way.
> First, for a non-dummy expert, $\phi$ maps it to another non-dummy expert optimally (to minimize the total loss after swap).
> Then, to define $\phi$'s behavior for dummy experts, we consider two cases:
> in the case where the algorithm picks dummy experts for more than $\sqrt{kT\log k}$ times, we let $\phi$ map all dummy experts to a fixed non-dummy expert (in which case $d_\phi^{\text{unif}} \geq d-k$); otherwise, all dummy experts are mapped to themselves (in which case $d_\phi^{\text{self}} \geq d-k$).
> Note that in both cases, we have $c_\phi \leq k+1$.
> Moreover, in the first case, $\mathrm{Reg}(\phi)$ is at least $\frac{1}{2}\sqrt{kT\log k}$ since whenever the algorithm picks a dummy expert $i$, it gets loss $1$ while expert $\phi(i)$ gets loss at most $1/2$ (and the rest of the regret is non-negative by construction).
> In the second case, $\mathrm{Reg}(\phi)$ is essentially the swap regret of the algorithm for a $k$-expert problem that lasts for at least $T-\sqrt{kT\log k}$ rounds,
> and since the instance comes from the lower bound construction of Ito [2020], we also have $\mathrm{Reg}(\phi)=\Omega(\sqrt{k\log(k)(T-\sqrt{kT\log k})}) = \Omega(\sqrt{kT\log k})$.
>
>
> ---
> >Q2: Regarding the kernelized MWU algorithm (Section 4), the paper mentions an $\mathcal{O}(d^3)$ complexity per iteration. How does this complexity impact the algorithm’s scalability for large $d$? Are there practical optimizations considered to handle high-dimensional scenarios?
>
> First, we point out that in response to Review zKH9's comment, we have in fact further improved the time complexity of Algorithm 6 from $\mathcal{O}(d^3)$ to $\mathcal{O}(d^2)$ (see proof therein).
> So the final per-round complexity of our algorithm is $\mathcal{O}(d^2)+\textrm{FP}_d$ where $\textrm{FP}_d$ is the complexity of computing the fixed point of a $d\times d$ stochastic matrix, which is $\mathcal{O}(d^\omega)$ in theory with $\omega \approx 2.37$, but can be much faster in practice using e.g., power method, as also pointed out by Review zKH9.
> Importantly, our algorithm is thus computationally as efficient as the classic Blum-Mansour algorithm and also the Lu et al. [2025] algorithm (while enjoying strictly better and more adaptive regret bounds).
>
>
> ---
> >Q3: Can the framework be extended to handle heterogeneous agents with different action spaces and loss functions, as encountered in real-world games, and if so, how would the current analysis need to be adapted to accommodate such heterogeneity?
>
> First, we point out that we definitely do not require the same loss functions for different agents, since we are considering general-sum games.
> Second, as pointed out in our Footnote 4, we assume that the action set size is the same for all players, but that is completely for notational conciseness, and our analysis
> can be directly extended to games with different action set sizes, with $d$ now representing the maximum action set size among the agents.

---

> > ### Comment · Reviewer_WnBk · 2025-08-05
> >
> > Thank you for your rebuttal. It addressed my concerns and questions well. I think this is a technically solid paper, and I have therefore raised my score to Accept.

---

### Official Review · Reviewer_Szbw · 2025-07-02

**Clarity:** 3
**Significance:** 3
**Originality:** 3
**Rating:** 5
**Confidence:** 4

**Summary:**

This paper builds upon the recent work of Lu et al. (2025), providing $\Phi$-regret bounds that are based on a sparsity complexity measure $c_\phi$ that recovers the almost optimal no-external, no-internal, and no-swap regret guarantees. $\Phi$-regret is a well-established regret notion, where given a sequence of probabilities $p_1,\ldots,p_T \in \Delta_d$, it measures the difference between the experienced cost and the cost of the best transformation $\phi: \Delta_d \mapsto \Delta_d$,
$\mathcal{R}(\phi):=\sum_{t=1}^T \ell_t^\top \cdot p_t - \ell_t^\top \cdot \phi(p_t)$. Lu et al. (2025) provide an algorithm with $\mathcal{R}(\phi) := O\left(\sqrt{c_\phi(T + d) \log^3 d}\right)$, while the authors provide an improved bound of $\mathcal{R}(\phi) := O\left(\sqrt{c_\phi T \log d}\right)$ with a significantly simpler algorithm.

The basic technical idea of the authors is to introduce a prior probability distribution on the set of $0$–$1$ row-stochastic matrices and to establish $\Phi$-regret bounds by running the famous MWU on this set. The authors then use MWU as a meta-algorithm in order to select the optimal step size. The authors additionally show that, despite dealing with an exponentially large action set, their algorithm can be efficiently implemented using kernelization ideas.

In Section $5$, the authors introduce an alternative approach based on the Blum–Mansour reduction. Finally, in Section 6, the authors show how to extend their algorithm so as to achieve $\tilde{\mathcal{O}}(1/T)$ for approximate $\Phi$-equilibrium in a specific class of normal-form games.

**Questions:**

1. What are the limitations of extending your results for general sum normal form games?

2. Can you elaborate more on the class of games satisfying Assumption~6.3?

**Ethical Concerns:**

["NO or VERY MINOR ethics concerns only"]

**Final Justification:**

After the author-reviewer discussion I am very confident on the significance of the results provided by the paper. I thus recommend acceptance.

**Limitations:**

Yes

**Quality:**

3

**Strengths And Weaknesses:**

I think the paper presents solid theoretical results that will be of interest to the online learning community at NeurIPS. Beyond the improved regret guarantees over those of Lu et al., the proposed algorithms are both simple and elegant. The presentation of the paper is very clear, and the technical ideas are well-explained.

Perhaps the only result I found somewhat expected was the $\tilde{O}(1/T)$ convergence rate to $\Phi$-equilibrium, given the extensive recent literature on the subject. Despite the considered class has been previously considered to the literature, the assumption feels a bit artificial. At the same time, I believe that a similar result for general normal-form games would add value to the paper.

---

> ### Author Rebuttal · Authors · 2025-07-30
>
> Thank you for the very positive review. We address your questions below.
>
> >Q1: What are the limitations of extending your results for general sum normal form games?
>
> We use the key nonnegative-social-external-regret assumption to establish a small path-length bound $\mathcal{O}(N\log d)$ of the entire learning dynamic (see Theorem D.7).
> Without this assumption, in principle, we could also use the fact that swap regret is by definition nonnegative to establish another path-length bound, but it would be larger than $\mathcal{O}(N\log d)$ by some factors of poly$(d)$, making the final bound on
> $\mathrm{Reg}\_n(\phi)$  at least as large as the best existing bound for swap regret [Anagnostides et al., 2022a,b] even for a  $\phi$ with small $c\_\phi$ (and thus vacuous for our purpose).
>
> ---
>
> >Q2: Can you elaborate more on the class of games satisfying Assumption~6.3?
>
> While there is no obvious intuitive explanation for this assumption,
> as argued by Anagnostides et al. [2022c], this class of games is intricately connected with the admission of a minimax theorem.
> They show (in their Proposition A.10) that all the following standard and well-studied games satisfy this property:
> 1. Two-player zero-sum game;
> 2. Polymatrix zero-sum games;
> 3. Constant-sum Polymatrix games;
> 4. Strategically zero-sum games;
> 5. Polymatrix strategically zero-sum games;
> 6. Convex-concave (zero-sum) games;
> 7. Zero-sum games with objective $f(x,y)$ such that
>    $$
>         \min\_{x\in \mathcal{X}}\max\_{y\in \mathcal{Y}} f(x,y)=\max\_{y\in \mathcal{Y}} \min\_{x\in \mathcal{X}}f(x,y)
>    $$
> 8. Quasiconvex-quasiconcave (zero-sum) games;
> 9. Zero-sum stochastic games.

---

> ### Comment · Reviewer_Szbw · 2025-08-02
> **Reviewer's Response**
>
> Thank you very much for your prompt repsonse. I maintain my positive impression of the paper and I keep my score.

---

### Decision · Program_Chairs · 2025-09-17

**Decision:**

Accept (spotlight)

**Comment:**

This paper was met with unanimous enthusiasm from all four reviewers. The work introduces a new framework for achieving comparator-adaptive Phi-regret that is simpler, more efficient, and achieves tighter bounds than previous state-of-the-art methods. A key contribution is the extension of these results to game theory, providing the first adaptive and accelerated convergence rates to Phi-equilibria. The authors' rebuttal was comprehensive and successfully addressed all reviewer concerns, including a key point about computational complexity that led one reviewer to raise their score. The work is certainly strong enough for a spotlight presentation.